# Exact Fractional Inference via Re-Parametrization & Interpolation between Tree-Re-Weighted- and Belief Propagation- Algorithms

**Hamidreza Behjoo**  *hamidreza.behjoo@gmail.com*
*University of Arizona*

**Michael Chertkov**  *chertkov@arizona.edu*
*University of Arizona*

**Reviewed on OpenReview:** *https://openreview.net/forum?id=AWRpSgaNfc*

## Abstract

Computing the partition function, $Z$, of an Ising model over a graph of $N$ "spins" is most likely exponential in $N$. Efficient variational methods, such as Belief Propagation (BP) and Tree Re-Weighted (TRW) algorithms, compute $Z$ approximately by minimizing the respective (BP- or TRW-) free energy. We generalize the variational scheme by building a $\lambda$-fractional interpolation, $Z^{(\lambda)}$, where $\lambda = 0$ and $\lambda = 1$ correspond to TRW- and BP-approximations, respectively. This fractional scheme – coined Fractional Belief Propagation (FBP) – guarantees that in the attractive (ferromagnetic) case $Z^{(TRW)} \geq Z^{(\lambda)} \geq Z^{(BP)}$, and there exists a unique ("exact") $\lambda_*$ such that $Z = Z^{(\lambda_*)}$. Generalizing the re-parametrization approach of (Wainwright et al., 2001) and the loop series approach of (Chertkov & Chernyak, 2006a), we show how to express $Z$ as a product, $\forall \lambda: Z = Z^{(\lambda)}\tilde{Z}^{(\lambda)}$, where the multiplicative correction, $\tilde{Z}^{(\lambda)}$, is an expectation over a node-independent probability distribution built from node-wise fractional marginals. Our theoretical analysis is complemented by extensive experiments with models from Ising ensembles over planar and random graphs of medium and large sizes. Our empirical study yields a number of interesting observations, such as the ability to estimate $\tilde{Z}^{(\lambda)}$ with $O(N^{2::4})$ fractional samples and suppression of variation in $\lambda_*$ estimates with an increase in $N$ for instances from a particular random Ising ensemble, where $[2 :: 4]$ indicates a range from 2 to 4. We also verify and discuss the applicability of this approach to the problem of image de-noising. Based on these experiments and the theory, we conclude that both the computation of the partition function and the computation of marginals can be efficiently performed via FBP at the optimal value of $\lambda_*$.

## 1 Introduction

Graphical Models (GM) are a major tool of Machine Learning that allow expressing complex statistical correlations via graphs. Ising models are the most widespread GM for expressing correlations between binary variables associated with nodes of a graph, where the probability is factorized into a product of terms, each associated with an undirected edge of the graph. Many methods of inference and learning in GM are first tested on Ising models and then generalized, , for example, beyond binary and pair-wise assumptions.

In this manuscript, we focus on computing the normalization factor $Z$ (called the partition function), over the Ising models. The problem is known to be of sharp-P complexity, likely requiring computational efforts that are exponential in the size (number of nodes, $N$) of the graph (Welsh, 1991; Jerrum & Sinclair, 1993; Goldberg & Jerrum, 2015; Barahona, 1982). There are three general approximate methods to compute $Z$: (a) elimination of (summation over) the variables one-by-one (Dechter, 1999; Dechter & Rish, 2003; Liu & Ihler, 2011; Ahn et al., 2018); (b) the variational approach (Yedidia et al., 2001; 2005); (c) Monte Carlo

(MS) sampling (Andrieu et al., 2003). (See also reviews (Wainwright & Jordan, 2008; Chertkov, 2023) and references therein.) In this manuscript, we develop the latter two methods. We also pay special attention to providing and tightening approximation guarantees. We base our novel theory and algorithm on the provable upper bound for $Z$ associated with the so-called Tree Re-Weighted (TRW) variational approximation (Wainwright et al., 2003; 2005) and on the Belief Propagation (BP) variational approximation (Yedidia et al., 2001; 2005), which is known to provide a lower bound on $Z$ in the case of an attractive (ferromagnetic) Ising model (Ruozzi, 2012). Note that there are also additional upper bounds derived from log-determinant relaxations, wherein binary graphical models are relaxed to Gaussian graphical models on the same graph (Wainwright & Jordan, 2006; Ghaoui & Gueye, 2008). However, these bounds are generally considered to be loose, and to the best of our knowledge, there is no known method to effectively narrow the gap between these upper bounds and the exact partition function.

## 1.1 Relation to Prior Work

In addition to the aforementioned relations to foundational work on the variational approaches (Yedidia et al., 2001; 2005), MCMC approaches (Andrieu et al., 2003), and lower and upper variational bounds (Ruozzi, 2012; Wainwright et al., 2003), this manuscript also builds on recent results in other related areas, in particular:

- We extend the ideas of parameterized interpolation between BP (Yedidia et al., 2001; 2005) and TRW (Wainwright et al., 2003; 2005), in the spirit of fractional BP (Wiegerinck & Heskes, 2002; Chertkov & Yedidia, 2013), thus introducing a broader family of variational approximations.

- Expanding on the previous point, since our approach can be considered as an interpolation bridging the TRW and BP approximations for the GM's partition function, it seems appropriate to cite (Knoll et al., 2023), where another interpolation approach was unveiled. The interpolation discussed in (Knoll et al., 2023) is of an annealing type – it starts with the trivial (high-temperature) model where all components are independent, and then the potentials of the pair-wise model are tuned gradually, adjusting the scaling parameter from 0 to 1. At every incremental step, a BP algorithm is employed to track the fixed point, ensuring the method's precision and reliability. The primary objective of this approach was to surpass the conventional BP in both accuracy and convergence rates. In contrast, our approach is designed to ascertain the exact value of the partition function and marginals, thus setting a new standard for precision in the analysis of complex systems.

- We utilize and generalize re-parametrization (Wainwright et al., 2001), gauge transformation, and loop calculus (Chertkov & Chernyak, 2006a;b; Chertkov et al., 2020) techniques, as well as the combination of the two (Willsky et al., 2007).

- Our approach is also related to the development of MCMC techniques with polynomial guarantees, the so-called Fully Randomized Polynomial Schemes (FPRS), developed specifically for Ising models of specialized types, e.g., attractive (Jerrum & Sinclair, 1993) and zero-field, planar (Gómez et al., 2010; Ahn et al., 2016).

## 1.2 This Manuscript's Contribution

We introduce a fractional variational approximation that interpolates between the classical Tree Re-Weighted (TRW) and Belief Propagation (BP) cases. The fractional free energy, $\bar{F}^{(\lambda)} = -\log Z^{(\lambda)}$, defined as the negative logarithm of the fractional approximation to the exact partition function, $Z = \exp(-\bar{F})$, requires solving an optimization problem, which is achieved practically by running a fractional version of one of the standard message-passing algorithms. The parameter $\lambda \in [0, 1]$ interpolates between the $\lambda = 1$ and $\lambda = 0$ cases, corresponding to BP and TRW, respectively. The interpolation technique, particularly our focus on $\lambda_*$, which lies somewhere between $\lambda = 0$ and $\lambda = 1$ and for which $Z^{(\lambda_*)} = Z$, is novel – to the best of our knowledge, this is the first manuscript where the interpolation technique is discussed. Basic definitions, including problem formulation for the Ising models and variational formulation in terms of the node and edge beliefs (proxies for the respective exact marginal probabilities), are given in Section 2. Assuming that

the fractional message-passing algorithm converges, we study the dependence of the fractional free energy on the parameter $\lambda$ and the relationship between the exact value of the free energy (the negative logarithm of the exact partition function) and the fractional free energy. We report the following theoretical results:

- We show in Section 3 that under some mild asumptions $\bar{F}^{(\lambda)}$ is a continuous and monotone function of $\lambda$ (Theorem 3.1 proved in Appendix B), which is also concave in $\lambda$ (Theorem 3.2).

- Our main theoretical result, Theorem 4.1, presented in Section 4 and proven in Appendix C, states that the exact partition function can be expressed as a product of the variational free energy and a multiplicative correction, $Z = Z^{(\lambda)}\tilde{Z}^{(\lambda)}$. The latter multiplicative correction term, $\tilde{Z}^{(\lambda)}$, is stated as an explicit expectation of an expression over a well-defined "mean-field" probability distribution, where both the expression and the "mean-field" probability distribution are stated explicitly in terms of the fractional node and edge beliefs. We note that such a bridge between the exact partition function and the approximate partition function, known as the Loop Series/Calculus, was introduced in Chertkov & Chernyak (2006a;b) and elaborated upon in Willsky et al. (2007) for the case of the Bethe (Belief Propagation), where $\lambda = 1$. However, to the best of our knowledge, it has not been reported in the literature for any other values of $\lambda \in [0, 1]$ interpolating between BP and TRW, particularly for $\lambda = 0$ corresponding to TRW.

The theory is extended with experiments reported in Section 5. Here we show, in addition to confirming our theoretical statements (and thus validating our simulation settings), that:

- Evaluating $Z^{(\lambda)}\tilde{Z}^{(\lambda)}$ at different values of $\lambda$ and confirming that the result is independent of $\lambda$ suggests a novel approach to a reliable and efficient estimate of the exact $Z$ – the Fractional Message Passing Algorithm 1.

- Analyzing ensembles of the attractive Ising models over graphs of size $N$, we observe that fluctuations of the value of $\lambda_*$ within the ensemble, where $Z^{(\lambda_*)} = Z$, decrease dramatically with an increase in $N$. This observation suggests that estimating $\lambda_*$ for an instance from the ensemble allows efficient approximate evaluation of $Z$ for any other instance from the ensemble.

- Studying the importance-sampling procedure based on TRW BP, we empirically estimate $\tilde{Z}^{(\lambda)}$ and observe convergence with the number of samples. Analyzing the speed of convergence with the number of nodes, $N$, in the Ising model, our experiments show that the correct mean of $\tilde{Z}^{(\lambda_*)}$, evaluated at $\lambda = \lambda_*$, is recovered with $N^2$ samples. Additionally, the variance of the respective importance sampling is reduced beyond a pre-fixed $\mathcal{O}(1)$ tolerance with $N^4$ samples.

- Analysis of mixed Ising ensembles (where attractive and repulsive edges alternate) suggests that for instances with sufficiently many repulsive edges, finding, $\lambda_* \in [0, 1]$ may not be feasible.

- We demonstrate the effectiveness of our approach in the context of image denoising, as detailed in Section 5.7. This application highlights the practical utility of our theoretical developments in a real-world machine learning task. Furthermore, the denoising experiments support the conjecture that the optimal $\lambda_*$ is special not only because the Fractional Belief Propagation (FBP) at this value yields the exact partition function, but also because estimating marginal probabilities at $\lambda_*$ is significantly more accurate. The error in these estimates decreases as the size of the problem increases, underscoring the robustness and scalability of our method.

We have a brief discussion of conclusions and paths forward in Section 6.

## 2 Technical Preliminaries

### 2.1 Ising Models: the formulation

Graphical Models (GM) are the result of a marriage between probability theory and graph theory designed to express a class of high-dimensional probability distributions that factorize in terms of products of lower-dimensional factors. The Ising model is an exemplary GM defined over an undirected graph, $\mathcal{G} = (\mathcal{V}, \mathcal{E})$. The

Ising Model is stated in terms of binary variables, $x_a = \pm 1$, and singleton factors, $h_a \in \mathbb{R}$, associated with nodes of the graph, $a \in \mathcal{V}$, and pair-wise factors, $J_{ab} \in \mathbb{R}$, associated with edges of the graph, $\{a, b\} \in \mathcal{E}$. The probability distribution of the Ising model observing a state, $\boldsymbol{x} = (x_a | a \in \mathcal{V})$ is

$$p(\boldsymbol{x}|\boldsymbol{J}, \boldsymbol{h}) = \frac{\exp\left(-E\left(\boldsymbol{x}; \boldsymbol{J}, \boldsymbol{h}\right)\right)}{Z(\boldsymbol{J}, \boldsymbol{h})}, \quad Z(\boldsymbol{J}, \boldsymbol{h}) := \sum_{\boldsymbol{x} \in \{\pm 1\}^{|\mathcal{V}|}} \exp\left(-E\left(\boldsymbol{x}; \boldsymbol{J}, \boldsymbol{h}\right)\right), \tag{1}$$

$$E\left(\boldsymbol{x}; \boldsymbol{J}, \boldsymbol{h}\right) := \sum_{\{a,b\} \in \mathcal{E}} E_{ab}(x_a, x_b), \quad \forall \{a, b\} \in \mathcal{E} : \; E_{ab}(x_a, x_b) = -J_{ab} x_a x_b - h_a x_a / d_a + h_b x_b / d_b, \tag{2}$$

where $d_a$ is the degree of node $a$ in $\mathcal{G}$ and $\boldsymbol{J} := (J_{ab} | \{a, b\} \in \mathcal{E})$, $\boldsymbol{h} = (h_a | a \in \mathcal{V})$ are the pair-wise and singleton vectors, assumed given. $E(\boldsymbol{x}; \boldsymbol{J}, \boldsymbol{h})$ is the energy function and $Z(\boldsymbol{J}, \boldsymbol{h})$ is the partition function. Solving the Ising model inference problem means computing $Z$ which generally requires an effort that is exponential in $N = |\mathcal{V}|$.

## 2.2 Exact Variational Formulation

Exact variational approach to compute $Z$ consists in restating Eq. (1) in terms of the following Kullback-Leibler distance between $\exp(-E(\boldsymbol{x}; \boldsymbol{J}, \boldsymbol{h}))$ and a probability distribution, $\mathcal{B} : \{-1, 1\}^{|\mathcal{V}|} \to [0, 1]$, $\sum_{\boldsymbol{x}} \mathcal{B}(\boldsymbol{x}) = 1$, called belief:

$$\bar{F} = -\log Z = \min_{\mathcal{B}(\boldsymbol{x})} \sum_{\boldsymbol{x}} \left(E(\boldsymbol{x})\mathcal{B}(\boldsymbol{x}) - \mathcal{B}(\boldsymbol{x}) \log \mathcal{B}(\boldsymbol{x})\right), \tag{3}$$

where $\bar{F}$ is also called the free energy (following widely accepted physics terminology).

The exact variational formulation (3) is the starting point for approximate variational formulations, such as BP (Yedidia et al., 2005) and TRW (Wainwright & Jordan, 2008), stated solely in terms of the marginal beliefs associated with nodes and edges, respectively:

$$\forall a \in \mathcal{V}, \; \forall x_a : \; \mathcal{B}_a(x_a) := \sum_{\boldsymbol{x} \backslash x_a} \mathcal{B}(\boldsymbol{x}), \quad \forall \{a, b\} \in \mathcal{E}, \; \forall x_a, x_b : \; \mathcal{B}_{ab}(x_a, x_b) := \sum_{\boldsymbol{x} \backslash (x_a, x_b)} \mathcal{B}(\boldsymbol{x}). \tag{4}$$

Moreover, the *fractional* approach developed in this manuscript provides a variational formulation in terms of the marginal probabilities, generalizing (and, in fact, interpolating between) the respective BP and TRW approaches. Therefore, we now turn to stating the fractional variational formulation.

## 2.3 Fractional Variation Formulation

Let us introduce a fractional-, or $\lambda$- *reparametrization* of the belief (proxy for the probability distribution of $\boldsymbol{x}$)

$$\mathcal{B}^{(\lambda)}(\boldsymbol{x}) = \frac{\prod_{\{a,b\} \in \mathcal{E}} \left(\mathcal{B}_{ab}(x_a, x_b)\right)^{\rho_{ab}^{(\lambda)}}}{\prod_{a \in \mathcal{V}} \left(\mathcal{B}_a(x_a)\right)^{\sum_{b \sim a} \rho_{ab}^{(\lambda)} - 1}}, \tag{5}$$

where $b \sim a$ is a shorthand notation for $b \in \mathcal{V}$ such that, given $a \in \mathcal{V}$, $\{a, b\} \in \mathcal{E}$. Here in Eq. (5), $\rho_{ab}^{(\lambda)}$ is the $\lambda$-parameterized edge appearance probability

$$\rho_{ab}^{(\lambda)} = \rho_{ab} + \lambda(1 - \rho_{ab}), \quad (\lambda : 0 \to 1), \tag{6}$$

which is expressed via the $\lambda = 0$ edge appearance probability, $\rho_{ab}$, dependent on the weighted set of spanning trees, $\mathcal{T} := \{T\}$, of the graph according to the following TRW rules (Wainwright, 2002; Wainwright & Jordan, 2008):

$$\forall \{a, b\} \in \mathcal{V} : \; \rho_{ab} = \sum_{T \in \mathcal{T}, \text{ s.t. } \{a,b\} \in T} \rho_T, \quad \sum_{T \in \mathcal{T}} \rho_T = 1. \tag{7}$$

Several remarks are in order. First, $\lambda = 1$ corresponds to the case of BP. Then Eq. (5) is exact in the case of a tree graph, but it can also be considered as a (loopy) BP approximation in general. Second, and as mentioned above, $\lambda = 0$, corresponds to the case of TRW. Third, the newly introduced (joint) beliefs are not globally consistent, i.e. $\sum_{\boldsymbol{x}} \mathcal{B}^{(\lambda)}(\boldsymbol{x}) \neq 1$ for any $\lambda$, including the $\lambda = 0$ (TRW) and $\lambda = 1$ (BP) cases.

Substituting Eq. (5) into Eq. (3) we arrive at the following fractional approximation to the exact free energy stated as an optimization over all the node and edge marginal beliefs, $\mathcal{B} := (\mathcal{B}_{ab}(x_a, x_b)|\forall\{a, b\} \in \mathcal{E},\ x_a, x_b = \pm 1) \cup (\mathcal{B}_a(x_a)|\forall a \in \mathcal{V},\ x_a = \pm 1)$:

$$\bar{F}^{(\lambda)} := F^{(\lambda)}(\mathcal{B}^{(\lambda)}), \quad \mathcal{B}^{(\lambda)} := \arg\min_{\mathcal{B} \in \mathcal{D}} F^{(\lambda)}(\mathcal{B}), \tag{8}$$

$$F^{(\lambda)}(\mathcal{B}) := E(\mathcal{B}) - H^{(\lambda)}(\mathcal{B}), \quad E(\mathcal{B}) := \sum_{\{a,b\} \in \mathcal{E}} \sum_{x_a, x_b = \pm 1} E_{ab}(x_a, x_b) \mathcal{B}_{ab}(x_a, x_b),$$

$$H^{(\lambda)}(\mathcal{B}) := - \sum_{\{a,b\} \in \mathcal{E}} \rho_{ab}^{(\lambda)} \sum_{x_a, x_b = \pm 1} \mathcal{B}_{ab}(x_a, x_b) \log \mathcal{B}_{ab}(x_a, x_b) + \sum_{a \in \mathcal{V}} \left( \sum_{b \sim a} \rho_{ab}^{(\lambda)} - 1 \right) \sum_{x_a = \pm 1} \mathcal{B}_a(x_a) \log \mathcal{B}_a(x_a),$$

$$\mathcal{D} := \left( \mathcal{B} \left| \begin{array}{ll} \mathcal{B}_a(x_a) = \sum_{x_b = \pm 1} \mathcal{B}_{ab}(x_a, x_b), & \\ \quad \forall a \in \mathcal{V},\ \forall b \sim a,\ \forall x_a = \pm 1; & (a) \\ \sum_{x_a, x_b = \pm 1} \mathcal{B}_{ab}(x_a, x_b) = 1, & \\ \quad \forall \{a, b\} \in \mathcal{E}; & (b) \\ \mathcal{B}_{ab}(x_a, x_b) \geq 0, & \\ \quad \forall \{a, b\} \in \mathcal{E},\ \forall x_a, x_b = \pm 1. & (c) \end{array} \right. \right). \tag{9}$$

As discussed in Section 3 in detail, $\lambda = 0$ results in $Z^{(\lambda)}$ which upper bounds the exact $Z$, and $\lambda = 1$ results in a lower bound if the model is attractive.

The optimization over beliefs in Eq. (8) can be restated in the Lagrangian form (see Appendix A.1). Fixed points of the Lagrangian (potentially many) satisfy the so-called message-passing equations (see Appendix A.2). We will refer to the iterative algorithm that finds marginal probabilities $\mathcal{B}$ and respective message variables $\mu$ by solving these equations as the basic Fractional Belief Propagation (FBP) algorithm. Consistent with the equations from Appendices A.1 and A.2, the resulting messages can then be used to find the FBP approximation, $Z^{(\lambda)}$, for the partition function $Z$ as follows:

$$Z^{(\lambda)} = \exp\left(-\bar{F}^{(\lambda)}\right) = \exp\left(-F^{(\lambda)}(\mathcal{B}^{(\lambda)})\right) = \prod_{\{a,b\} \in \mathcal{E}} \left( \sum_{x_a, x_b} \exp\left(-\frac{E_{ab}(x_a, x_b)}{\rho_{ab}^{(\lambda)}}\right) \times \tag{10}$$

$$\left(\mu_{b \to a}^{(\lambda)}(x_a)\right)^{\frac{\sum_{c \sim a} \rho_{ac}^{(\lambda)} - 1}{\rho_{ab}^{(\lambda)}}} \left(\mu_{a \to b}^{(\lambda)}(x_b)\right)^{\frac{\sum_{c \sim b} \rho_{bc}^{(\lambda)} - 1}{\rho_{ab}^{(\lambda)}}} \right)^{\rho_{ab}^{(\lambda)}} \prod_{a \in \mathcal{V}} \left( \sum_{x_a} \prod_{b \sim a} \mu_{b \to a}^{(\lambda)}(x_a) \right)^{1 - \sum_{c \sim a} \rho_{ac}^{(\lambda)}},$$

where $\mathcal{B}^{(\lambda)}$ is defined in Eq. (8) and its components are expressed via $\mu^{(\lambda)}$ and $\rho^\lambda$ according to Eqs. (17,18).

## 3 Properties of the Fractional Free Energy

Given the construction of the fractional free energy, described above in Section 2.3 and also detailed in Appendix A, we are ready to make the following statements.

**Theorem 3.1.** [Monotonicity of the Fractional Free Energy] Assuming $\boldsymbol{\rho} := (\rho_{ab}|\{a, b\} \in \mathcal{E})$ is fixed, and $\mathcal{B}^{(\lambda)}$ is differentiable in $\lambda$, $\bar{F}^{(\lambda)}$ is a continuous, monotone function of $\lambda$.

*Proof.* See Appendix B. □

*Remark.* The continuity of $\mathcal{B}^{(\lambda)}$ in $\lambda$ is a technical condition that almost always holds. This condition may break, albeit in some rare cases, when at a particular value of $\lambda$, two (or more) distinct local minima of the fractional free energy $F^{(\lambda)}(\mathcal{B})$ (considered as a function of $\mathcal{B}$) yield the same value. In such instances, as we vary $\lambda$ and pass through this special value of $\lambda$, the absolute minimum can jump from one local minimum to another.

**Theorem 3.2.** [Concavity of the Fractional Free Energy] Assuming $\boldsymbol{\rho} := (\rho_{ab}|\{a,b\} \in \mathcal{E})$ is fixed, $\bar{F}^{(\lambda)}$ is a concave function of $\lambda$.

*Proof.* Since $\rho_{ab}^{(\lambda)}$ is linear in $\lambda$, from Eq. (9), $H^{(\lambda)}(\mathcal{B})$ is also linear in $\lambda$. Therefore, for any $\lambda$, $\lambda_0$, $\lambda_1$ with $\lambda_0 < \lambda_1$, we have

$$H^{(\lambda)}(\mathcal{B}) = \frac{\lambda_1 - \lambda}{\lambda_1 - \lambda_0} H^{(\lambda_0)}(\mathcal{B}) + \frac{\lambda - \lambda_0}{\lambda_1 - \lambda_0} H^{(\lambda_1)}(\mathcal{B}),$$

and consequently,

$$F^{(\lambda)}(\mathcal{B}) = \frac{\lambda_1 - \lambda}{\lambda_1 - \lambda_0} F^{(\lambda_0)}(\mathcal{B}) + \frac{\lambda - \lambda_0}{\lambda_1 - \lambda_0} F^{(\lambda_1)}(\mathcal{B}).$$

Thus, we have

$$
\begin{aligned}
\bar{F}^{(\lambda)} = F^{(\lambda)}(\mathcal{B}^{(\lambda)}) &= \frac{\lambda_1 - \lambda}{\lambda_1 - \lambda_0} F^{(\lambda_0)}(\mathcal{B}^{(\lambda)}) + \frac{\lambda - \lambda_0}{\lambda_1 - \lambda_0} F^{(\lambda_1)}(\mathcal{B}^{(\lambda)}) \\
&\geq \frac{\lambda_1 - \lambda}{\lambda_1 - \lambda_0} F^{(\lambda_0)}(\mathcal{B}^{(\lambda_0)}) + \frac{\lambda - \lambda_0}{\lambda_1 - \lambda_0} F^{(\lambda_1)}(\mathcal{B}^{(\lambda_1)}) \\
&= \frac{\lambda_1 - \lambda}{\lambda_1 - \lambda_0} \bar{F}^{(\lambda_0)} + \frac{\lambda - \lambda_0}{\lambda_1 - \lambda_0} \bar{F}^{(\lambda_1)},
\end{aligned}
$$

where we used the fact that $F^{(\lambda)}(\mathcal{B}^{(\lambda)}) = \min_{\mathcal{B}} F^{(\lambda)}(\mathcal{B}) \leq F^{(\lambda)}(\mathcal{B}^{(\lambda')})$ at the inequality step. This proves the concavity of $\bar{F}^{(\lambda)}$ in $\lambda$. $\square$

Note that all the statements in this manuscript so far are made for arbitrary Ising models, i.e., without any restrictions on the graph and vectors of the pair-wise interactions, $\boldsymbol{J}$, and singleton biases, $\boldsymbol{h}$. If the discussion is limited to attractive (ferromagnetic) Ising models, $\forall\{a,b\} \in \mathcal{E} : J_{ab} \geq 0$, the following statement becomes a corollary of Theorem 3.1:

**Lemma 3.3.** [Exact Fractional] In the case of an attractive Ising model, any fixed $\boldsymbol{\rho}$, and also assuming that $\mathcal{B}^{(\lambda)}$ is differentiable in $\lambda$, there exists $\lambda_* \in [0,1]$ such that $Z^{(\lambda_*)} = Z$.

*Proof.* In the case of attractive Ising model, BP provides a lower bound on the exact partition function, $Z^{(\lambda=1)} \leq Z$, as proven in (Ruozzi, 2012). On the other hand, we know from (Wainwright & Jordan, 2008), and also by construction, that $Z^{(\lambda=0)} \geq Z$, i.e., the TRW estimate of the partition function provides an upper bound to the exact partition function. These lower and upper bounds, combined with the monotonicity of $Z^{(\lambda)}$ stated in Theorem 3.1, result in the desired statement. $\square$

## 4 Fractional Re-Parametrization for Exact Inference

**Theorem 4.1.** [Exact Relation Between $Z$ and $Z^{(\lambda)}$ for any $\lambda \in [0,1]$ ],

$$Z = Z^{(\lambda)} \tilde{Z}^{(\lambda)}, \tag{11}$$

$$\tilde{Z}^{(\lambda)} := \sum_{\boldsymbol{x}} \frac{\prod_{\{a,b\} \in \mathcal{E}} \left( \mathcal{B}_{ab}^{(\lambda)}(x_a, x_b) \right)^{\rho_{ab}^{(\lambda)}}}{\prod_{a \in \mathcal{V}} \left( \mathcal{B}_a^{(\lambda)}(x_a) \right)^{\sum_{c \sim a} \rho_{ac}^{(\lambda)} - 1}} = \mathbb{E}_{\boldsymbol{x} \sim p_0^{(\lambda)}(\cdot)} \left[ \frac{\prod_{\{a,b\} \in \mathcal{E}} \left( \mathcal{B}_{ab}^{(\lambda)}(x_a, x_b) \right)^{\rho_{ab}^{(\lambda)}}}{\prod_{a \in \mathcal{V}} \left( \mathcal{B}_a^{(\lambda)}(x_a) \right)^{\sum_{c \sim a} \rho_{ac}^{(\lambda)}}} \right], \tag{12}$$

where $p_0^{(\lambda)}(\boldsymbol{x}) := \prod_a \mathcal{B}_a^{(\lambda)}(x_a)$ is the component-independent distribution devised from the FBP-optimal node-marginal probabilities.

*Proof.* See Appendix C. $\square$

---

**Algorithm 1:** $\lambda$-optimal Fractional Belief Propagation

---

**Input:** $\mathcal{G} = (\mathcal{V}, \mathcal{E})$, graph.
**Initialize:** $\rho_{ab} = (|\mathcal{V}| - 1)/|\mathcal{E}|$
**For:** $\lambda = 0 : 0.05 : 1$,

     1. Compute $\rho_{ab}^{(\lambda)} = \rho_{ab} + \lambda(1 - \rho_{ab})$.

     2. Use Eq. (10) (and formulas from Appendix A.2) to find $Z^{(\lambda)}, \mathcal{B}_a^{(\lambda)}(x_a), \mathcal{B}_{ab}^{(\lambda)}(x_a, x_b)$

     3. Compute $\tilde{Z}^{(\lambda)}$ utilizing Eq. (12)

**End**

     1. Find $\lambda_*$ where $\tilde{Z}^{(\lambda)} = 1$

     2. Return $Z = Z^{(\lambda_*)}$

---

Notice that $\tilde{Z}^{(\lambda)}$, defined in Eq. (12), is the exact multiplicative correction term expressed in terms of the FBP solution. This term should equal 1 at the optimal value of $\lambda^*(\boldsymbol{J}, \boldsymbol{h})$. According to Lemma 3.3, this optimal value is achievable in the case of the attractive Ising model.

Theorem 4.1 suggests using Algorithm 1, presented as a pseudo-algorithm, which we refer to as the $\lambda$-optimal (or simply optimal) Fractional Belief Propagation Algorithm (O-FBP), to approximate the exact partition function $Z$.

We will also explore an alternative method for transforming the main theoretical result of this paper, Theorem 4.1, into a practical computational tool in the next subsection.

### 4.1 Optimal $\lambda$ from $Z$

---

**Algorithm 2:** Optimal $\lambda$ from $Z$

---

**Input:** $\mathcal{G} = (\mathcal{V}, \mathcal{E})$, graph. Exact value of $Z$.
**Initialize:** $\forall \{a, b\} \in \mathcal{E} : \rho_{ab} = (|\mathcal{V}| - 1)/|\mathcal{E}|$
**For:** $\lambda = 0 : 0.05 : 1$,

     1. Compute $\forall \{a, b\} \in \mathcal{E} : \rho_{ab}^{(\lambda)} = \rho_{ab} + \lambda(1 - \rho_{ab})$.

     2. Use Eq. (10) (and formulas from Appendix A.2) to find $Z^{(\lambda)}$

**End**

     1. Find $\lambda_*$ where $Z^{(\lambda)} = Z$

     2. Return $Z = Z^{(\lambda_*)}$

---

Suppose we have access to the exact value of the partition function $Z$. Then, Theorem 4.1 suggests Algorithm 2, which allows us to identify the optimal $\lambda_*$, that is, the value of $\lambda$ for which FBP (which can be evaluated efficiently, typically in linear time with respect to the problem size) outputs the exact result for the partition function. The exact value of $Z$, required as an input for Algorithm 2, can be determined via the following TRW- (or BP-) based computations:

     1. Compute $Z^{(0)}$ (or $Z^{(1)}$).

     2. Compute $\tilde{Z}^{(0)}$ (or $\tilde{Z}^{(1)}$) by sampling according to Eq. (10) (and the formulas in Appendix A.2).

     3. Use the identity $Z = Z^{(1)}\tilde{Z}^{(1)}$ (or $Z = Z^{(0)}\tilde{Z}^{(0)}$).

As we will see in the next section, where we turn to the empirical exploration of the approach presented in this paper, Algorithm 2 proves to be useful for making efficient computations of marginals in high-dimensional case.

## 5 Numerical Experiments

### 5.1 Setting, Use Cases and Methodology

In this section, we present the results of our numerical experiments, supporting and also further developing the theoretical results of the preceding sections. Specifically, we will describe the details of our experiments with the Ising model in the following "use cases:" (1) Over an exemplary planar graph – $N \times N$ square grid, where $N = [3 :: 5]$; (2) Over a fully connected graph, $K_N$, where $N = [9 :: 5^2]$. The notation $[a :: b]$ indicates a range from $a$ to $b$.

In both cases, we consider attractive models and mixed models – that is, models with some interactions being attractive (ferromagnetic), $J_{ab} > 0$, and some repulsive (antiferromagnetic), $J_{ab} < 0$. We experiment with the zero-field case, $\boldsymbol{h} = 0$, and also with the general (non-zero field) case. All of our models are "disordered" in the sense that we have generated samples of random $\boldsymbol{J}$ and $\boldsymbol{h}$. Specifically, in the attractive (mixed) case, components of $\boldsymbol{J}$ are i.i.d. from the uniform distribution, $\mathcal{U}(0, 1)$ ($\mathcal{U}(-1, 1)$), and components of $\boldsymbol{h}$ are i.i.d. from $\mathcal{U}(-1, 1)$. In some of our experiments, we draw a single instance of $\boldsymbol{J}$ and $\boldsymbol{h}$ from the respective ensemble. However, in other experiments – aimed at analyzing the variability within the respective ensemble – we show results for a number of instances.

We acknowledge that there is significant flexibility in selecting a set of spanning trees and then re-weighting respective contributions to $\boldsymbol{\rho} := (\rho_{ab} \mid \{a, b\} \in \mathcal{E})$ according to Eq. (7). (See some discussion of experiments with possible $\boldsymbol{\rho}$ in (Wainwright et al., 2005).) However, we chose not to test this flexibility. Instead, in all of our experiments, $\boldsymbol{\rho}$ is chosen uniformly for a given graph. A direct corollary of Lemma 7.3.2 from (Wainwright, 2002) is that if the edge-uniform re-weighting is feasible, the following relation holds for all $a, b \in \mathcal{E}$: $\rho_{ab} = (|\mathcal{V}| - 1)/|\mathcal{E}|$. This lemma allows us to restate the set of linear constraints defining the polytope in the TRW rules of Eq. (7) solely in terms of the edge-weights, $\rho_{ab}$, thereby excluding tree-weights. Moreover, it was shown in (Wainwright, 2002) that the edge-uniform re-weighting is feasible and optimal, providing the lowest TRW upper-bound in the case of highly symmetric graphs, such as fully connected or double-periodic square grids. Our experiments are conducted on graphs where identifying the feasibility of edge-uniform re-weighting is straightforward. However, we also note, consistently with a remark in (Wainwright, 2002), that there exist some exotic graphs for which edge-uniform assignment is infeasible.

We introduced the $\lambda$-optimal FBP Algorithm 1, which approximately calculates the partition function for a specified Ising model on a graph $\mathcal{G} = (\mathcal{V}, \mathcal{E})$. This algorithm generalized traditional message-passing methods by interpolating between the Tree-Reweighted case ($\lambda = 0$) and the Belief Propagation case ($\lambda = 1$) for any $\lambda \in [0, 1]$. Utilizing Theorem 4.1, the algorithm identifies a particular value, denoted as $\lambda_*$, where the fractional partition function $Z^{(\lambda_*)}$ is approximately equal to the exact partition function $Z$. The algorithm employs Eq. (12) to determine the correction factor $\tilde{Z}^{(\lambda)}$ utilizing fractional node and edge beliefs. Additionally, we initialize the edge appearance probabilities $\rho_{ab}$ uniformly.

To compute the fractional free energy, $\bar{F}^{(\lambda)}$ (minus the log of the fractional estimate for the partition function), we generalize the approach of (Bixler & Huang, 2018), which allows efficient, sparse-matrix-based implementation. Our code is available on `https://github.com/hamidrezabehjoo/Fractional-TRW`.

To compare the fractional estimate $\bar{F}^{(\lambda)} = -\log Z^{(\lambda)}$ with the exact free energy, $\bar{F} = -\log Z$, we either use direct computations (feasible for the $8 \times 8$ grid or smaller and for the fully connected graph over 64 nodes or smaller) or, in the case of the planar grid and zero-field, when computation of the partition function is reduced to computing a determinant, we use the code from (Likhosherstov et al., 2019) (see also references therein). Our computations are done for values of $\lambda$ equally spaced with the increment 0.05, between 0 and 1.

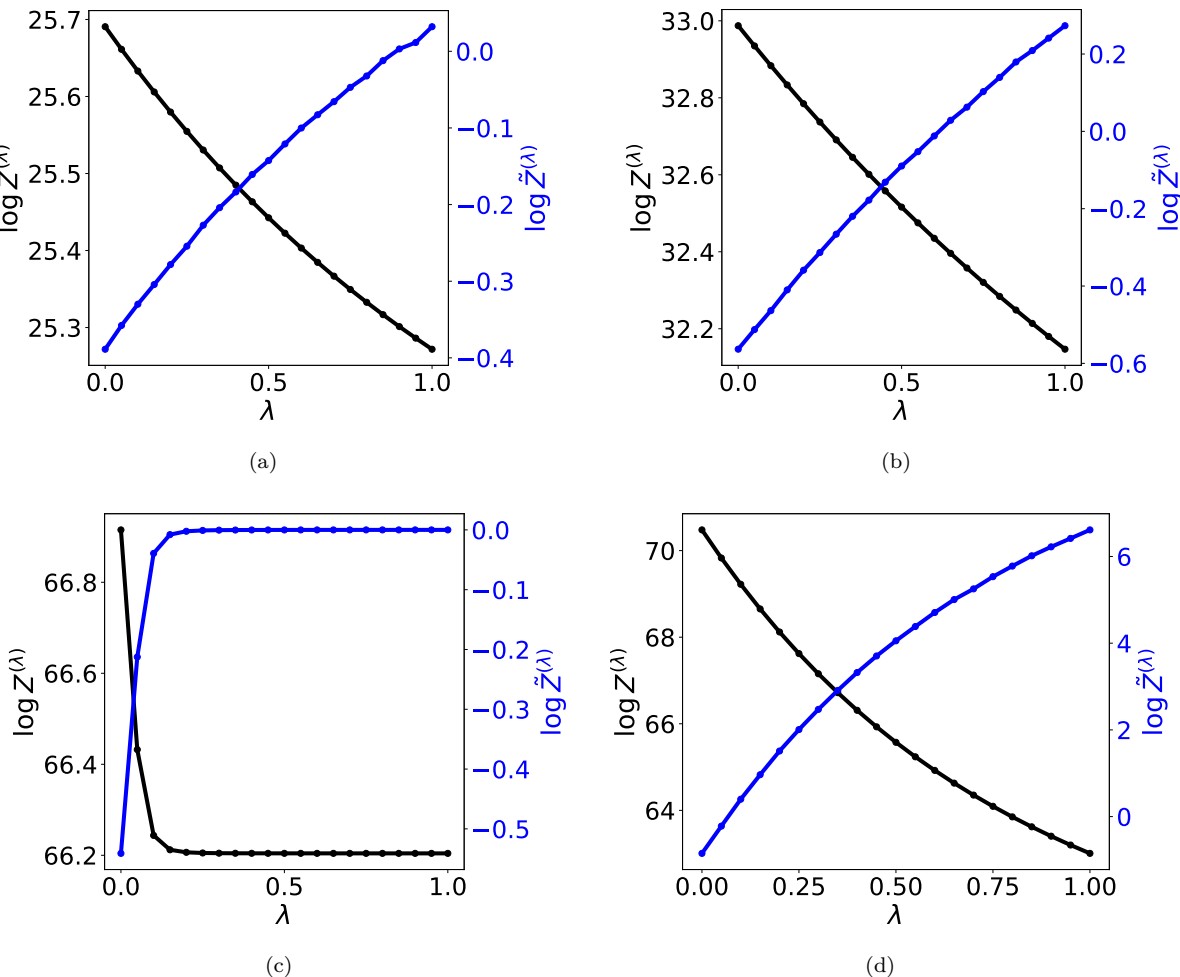

Figure 1: The case of the attractive Ising Model (a) with non-zero field and random interaction, $\boldsymbol{h}, \boldsymbol{J} \sim \mathcal{U}(0,1)$ on a $3 \times 3$ planar grid; and (b) with zero field and random interaction, $\boldsymbol{J} \sim \mathcal{U}(0,1)$ on a $3 \times 3$ planar grid; (c) with non-zero field and random interaction, $\boldsymbol{h}, \boldsymbol{J} \sim \mathcal{U}(0,1)$ on a $K_9$ complete graph and (d) with zero field and random interaction, $\boldsymbol{J} \sim \mathcal{U}(0,1)$ on a $K_9$ complete graph. We show fractional log-partition function (minus fractional free energy) on the left and the respective correction factor $\tilde{Z}^{(\lambda)}$ on the right vs the fractional parameter, $\lambda$. We observe monotonicity and concavity of $\bar{F}^{(\lambda)}$ on $\lambda$.

The log-correction term, $\log \tilde{Z}^{(\lambda)} = \log Z - \log Z^{(\lambda)}$, is estimated by direct sampling according to Eq. (12). (See Fig. 3 and the respective discussion below for empirical analysis of the number of samples required to guarantee sufficient accuracy.)

It is important to stress that, even though the $\lambda$-optimal FBP Algorithm 1 is a direct extension of what was discussed in the literature in the past for the TRW $\lambda = 0$ and BP $\lambda = 1$ cases, extending the algorithm to the interpolating $\lambda \in [0, 1]$ values is novel. In this regard, the $\lambda = 0$ and $\lambda = 1$ versions of the $\lambda$-optimal FBP Algorithm should be considered as providing baselines/benchmarks for its performance at the interpolating values of $\lambda$.

## 5.2 Properties of the Fractional Free Energy

We use Algorithm 1 for the fractional estimate of the log-partition function (minus fractional free energy), $\log Z^{(\lambda)} = -\bar{F}^{(\lambda)}$, and the log of the correction term, $\log \tilde{Z}^{(\lambda)} = \log Z - \log Z^{(\lambda)} = \bar{F}^{(\lambda)} - \bar{F}$. The results are

shown as functions of $\lambda$ in Fig. 1 where the monotonicity and concavity of $\bar{F}^{(\lambda)}$, proven in Theorem 3.1 and Theorem 3.2, respectively, are confirmed.

### 5.3 Relation between Exact and Fractional

Figs. 1, also provide empirical confirmation of Lemma 3.3 on existence of $\lambda_*$ in the case of attractive Ising model for which $Z^{(\lambda_*)} = Z$. Moreover, the full statement of Theorem 4.1, i.e., the equality between the left- and right-hand sides of Eq. (11), is also confirmed in all of our simulations (over Ising models, attractive or not) with high accuracy (when we can verify it by computing $Z$ directly).

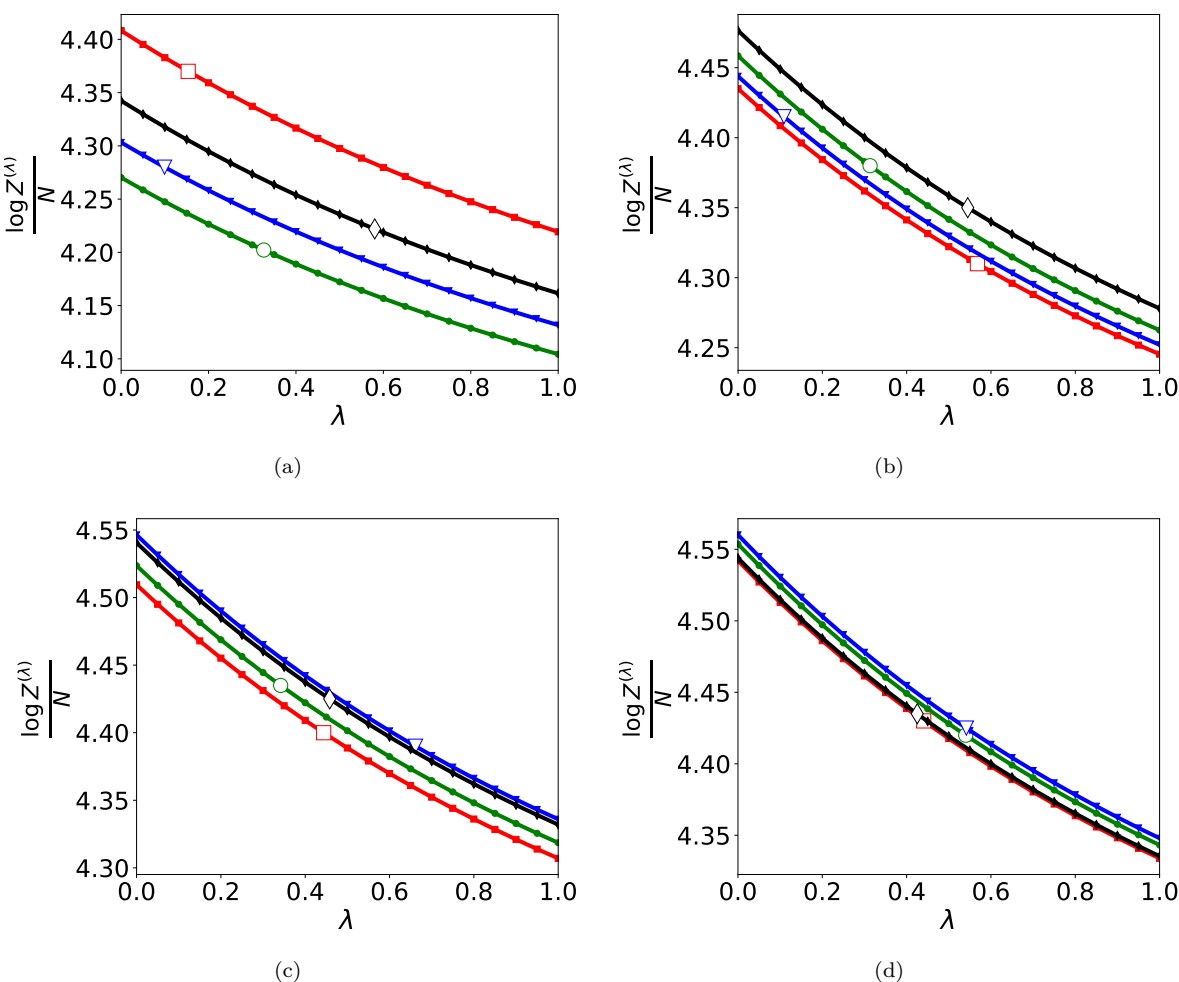

Figure 2: Planar zero-field Ising models for $n \times n$ grid with $J \sim \mathcal{U}(0,1)$. For each $n$, four different instances are generated by sampling uniformly at random from the unit interval and the exact values, $\lambda_*$, are shown by open symbols on each graph. (a) $10 \times 10$ (b) $20 \times 20$ (c) $30 \times 30$ (d) $40 \times 40$.

### 5.4 Concentration of the Fractional Parameter in Large Ensembles

Fig. 5 in Appendix D shows the dependence of $\bar{F}^{(\lambda)}$ on the fractional parameter, $\lambda$, for a number of instances drawn from two exemplary attractive use-case ensembles. We observe that the variability in the value of $\bar{F}^{(\lambda)}$ is significant. Variability in $\lambda_*$, where $Z^{(\lambda_*)} = Z$, is also observed, even though it is significantly smaller.

This observation suggests that the variability of $\lambda_*$ within an attractive ensemble decreases as $N$ grows. This hypothesis is confirmed in our experiments with larger attractive ensembles, illustrated in Fig. 2 for

different $N$. For each $N$ in the case of an $N \times N$ grid, we generate 4 different instances. We observe that as $N$ increases, the variability of $\lambda_*$ within the ensemble decreases dramatically. This observation is quite remarkable, as it suggests that it is sufficient to estimate $\lambda_*$ for one instance in a large ensemble and then use it for accurate estimation of $Z$ by simply computing $Z^{(\lambda_*)}$. This is also indicated in Algorithm 3. Our estimations, based on the data shown in Fig. 2, suggest that the width of the probability distribution of $\lambda_*$ within the ensemble scales as $\propto 1/\sqrt{N}$ with an increase in $N$.

---

**Algorithm 3:** Fractional Belief Propagation for Large Ensembles

**Input:** Ensemble $\mathcal{G}$ of similar graphical models, number of samples $M$, new instance $G_{new}$
**Output:** Estimated partition function $\hat{Z}$ and marginals $\{\hat{\mathcal{B}}_i\}, \{\hat{\mathcal{B}}_{ij}\}$ for $G_{new}$
**Function** PrecomputeLambdaStar($\mathcal{G}, M$)**:**

    $\Lambda \leftarrow \emptyset$;
    **for** $i = 1$ **to** $M$ **do**
        $G_i \sim \mathcal{G}$ ;               // Sample a graphical model from the ensemble
        $\lambda_i^* \leftarrow \text{FindOptimalLambda}(G_i)$ ;           // Using Algorithm 1
        $\Lambda \leftarrow \Lambda \cup \{\lambda_i^*\}$;
    **end**
    **return** $\bar{\lambda}^* = \frac{1}{M} \sum_{\lambda \in \Lambda} \lambda$ ;           // Average $\lambda^*$
**Function** ComputePartitionAndMarginals($G_{new}, \bar{\lambda}^*$)**:**

    Compute edge appearance probabilities $\rho_{ij}^{(\bar{\lambda}^*)}$ using Eq. (6);
    Run TRW algorithm with $\rho_{ij}^{(\bar{\lambda}^*)}$ on $G_{new}$;
    Obtain $Z^{(\bar{\lambda}^*)}$ using Eq. (10); and marginals $\{\hat{\mathcal{B}}_i\}, \{\hat{\mathcal{B}}_{ij}\}$;
    **return** $Z^{(\bar{\lambda}^*)}, \{B_i^{(\bar{\lambda}^*)}\}, \{B_{ij}^{(\bar{\lambda}^*)}\}$;

---

## 5.5 Convergence of Sampling for Fractional Partition Function

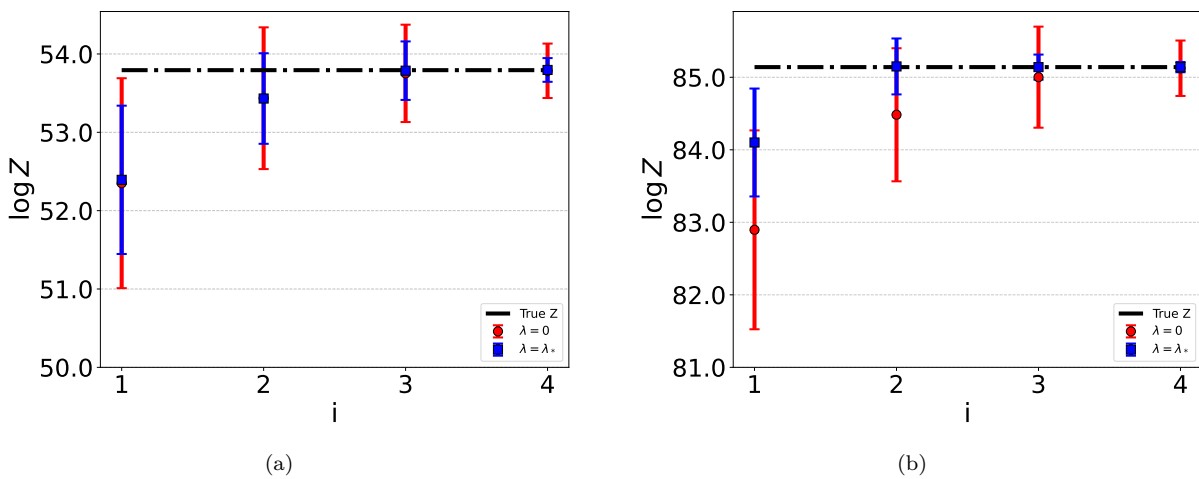

(a)                        (b)

Figure 3: Dependence of the sample-based estimate of $Z$ on the number of samples in the case of attractive Ising model for two different values of $\lambda$ in the case of (a) $4 \times 4$, and (b) $5 \times 5$ grids, where elements of $\boldsymbol{J}$ and $\boldsymbol{h}$ are drawn i.i.d. from $\mathcal{U}[0,1]$. The number of samples are $N^i$, where $i = 1, 2, 3, 4$ and $N = |\mathcal{V}|$ (that is $N = 16$ in (a) and $N = 25$ in (b).

Fig. 3 shows the dependence of the importance sampling-based estimate of $Z$ on the number of samples and $\lambda$, where the experiment was performed 100 times, and we report the mean and variance. Our major observation here is that the result converges with an increase in the number of samples. Moreover, comparing

the speed of convergence with the size of the system, $N$, we conjecture that the number of samples needed for the convergence of the mean scales as $\mathcal{O}(N^4)$ in the case of a generic $\lambda$. However, the scaling becomes much better, $\mathcal{O}(N^2)$, at the optimal $\lambda_*$. Another empirical conclusion extracted from Fig. 3 is that the variance of the importance sampling for estimating $\tilde{Z}^{(\lambda)}$ at $\lambda = \lambda_*$ reaches an $\mathcal{O}(1)$ (predefined) tolerance with the $N^4$ samples.

### 5.6 Fractional Approach for Mixed (Attractive and Repulsive) Cases

Fig. 6 in Appendix D shows two distinct situations which may be observed in the mixed case where some of the interactions are attractive but others are repulsive, allowing $Z^{(\lambda)}$ to be smaller or larger than $Z$. The former case is akin to the attractive model and $\lambda_* \in [0, 1]$, while in the latter case there exists no $\lambda_* \in [0, 1]$ such that $Z^{(\lambda_*)} = Z$.

### 5.7 Application in Machine Learning – Image De-Noising

Consider a black-and-white image represented as a binary vector, $\boldsymbol{x} = (\boldsymbol{x}_a = \pm 1 | a \in \mathcal{V})$, where $\mathcal{V}$ is the set of nodes in a two-dimensional $n \times n$ square grid. For example, the cameraman image shown in Fig. 4 is a $256 \times 256 = 65536$ pixel image, which is represented as a binary vector, $\boldsymbol{x} \in \{\pm 1\}^{65536}$.

The de-noising problem is set up as follows Koller & Friedman (2009): assume that an image is sent through a noisy Bernoulli channel, where each pixel is flipped independently with a probability $\varepsilon$. The noisy version of the image $\boldsymbol{y}$ is received, and the task is to recover the original image $\boldsymbol{x}$.

We also make the plausible assumption that images are constructed in such a way that the probability for two neighboring pixels $a$ and $b$ to have the same values $x_a$ and $x_b$, $\exp(J)/(\cosh(J))$, is higher than the probability $\exp(-J)/(\cosh(J))$ that they have opposite values, where $J > 0$.

Then, the probability for the image $\boldsymbol{x}$ to be reconstructed from the observed noisy image $\boldsymbol{y}$ is given by

$$p(\boldsymbol{x}|\boldsymbol{y}) \propto \exp\left(J \sum_{\{a,b\} \in \mathcal{E}} x_a x_b + h \sum_{a \in \mathcal{V}} x_a y_a\right), \quad h = \frac{1}{2} \log\left(\frac{\varepsilon}{1 - \varepsilon}\right), \tag{13}$$

where $\mathcal{G} = (\mathcal{V}, \mathcal{E})$ is the graph of the two-dimensional grid, and $\mathcal{E}$ is the set of edges of the grid. Clearly, Eq. (13) shows the probability distribution of the ferromagnetic Ising model.

The basic FBP algorithm solving Eq. (8), as well as its BP and TRW versions, with $\lambda = 1$ and $\lambda = 0$ respectively, can all be utilized to solve the de-noising problem.

Results of experiments de-noising an image with different algorithms are shown in Fig. 4, where pixels are flipped with the noise level corresponding to $h = 1.1$. We then optimize the $J$ parameter in each algorithm to obtain the best performance, evaluated according to the following error function:

$$\text{Error} = \frac{\text{Number of pixels different in true image and denoised version}}{\text{Total number of pixels}}.$$

We find that in the BP and TRW cases, the optimal $J$ values (minimizing the error) are 0.28 and 0.32 respectively. In the case of the basic FBP, we optimize not only over $J$ but also over $\lambda$, resulting in optimal values of $J = 0.3$ and $\lambda = 0.1$. We observe that the basic FBP algorithm demonstrates improved performance compared to both BP and TRW algorithms.

Note that this example is too large to reliably evaluate our $\lambda$-optimal FBP algorithm 1. However, we conjecture that the value of $\lambda$ obtained by minimizing the error will converge, in the thermodynamic limit of a large graph, to the value of $\lambda$ which is optimal within Algorithm 1. This conjecture is intuitively based on two facts. First, the error is expressed in terms of pixel marginals. Second, the $\lambda$ that optimizes the pixel marginals should be close to the $\lambda$ that optimizes the partition function. This is because, in the thermodynamic limit, the marginals can be reformulated in terms of the partition functions of the graphical models, which differ from the original one only by fixing the respective $x_a$ variable (to $\pm 1$) at a single pixel among many.

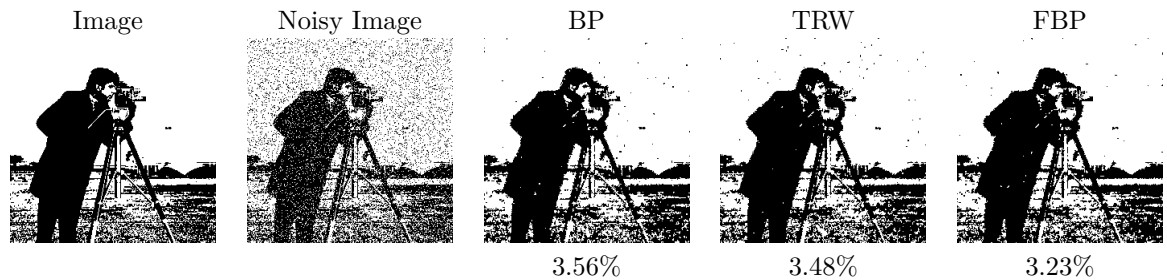

Figure 4: Different algorithms and their corresponding errors (listed below each image) for image de-noising.

It is also important to emphasize that even though the optimal $\lambda_*$ was defined for $Z$ – the partition function – inspired by results reported in Section 5.5, we have used it in this section to compute marginals. Indeed, a major empirical observation made in Section 5.5 suggests that computing the multiplicative correction $\tilde{Z}$ by sampling at the optimal value, $\lambda_*$, requires significantly fewer samples. In other words, the variance of sampling is minimal at the optimal value. This suggests that evaluating other expectations, particularly those corresponding to marginals, will also require fewer samples. Moreover, we also observed that the variance decreases with the model size. Based on all these observations, as well as the empirical advantage of evaluating marginals via FBP at $\lambda_*$ illustrated in Fig. 4, we hypothesize that evaluating FBP at the optimal $\lambda_*$ will provide exact results not only for the partition function but also asymptotically (when the problem size increases) for marginals.

Adding to this discussion, observations reported in Section 5.4 on the concentration of $\lambda_*$ in large ensembles provide further support for the hypothesis that it is sufficient to estimate $\lambda_*$ for only one (representative) GM from the ensemble. This result can then be used not only to estimate the partition function in other GM models of the ensemble but also to compute marginals via FBP efficiently. Note that this hypothesis also emphasizes the utility of Algorithm 2, which allows us to estimate $\lambda_*$ from the exact $Z$.

## 6 Conclusions and Path Forward

This manuscript suggests a new promising approach to evaluating inference in Ising Models. The approach consists in, first, solving a fractional variational problem via a distributed algorithm resulting in the fractional estimations for the partition function and marginal beliefs. We then compute multiplicative correction to the fractional partition function by evaluating a well-defined expectation of the mean-field probability distribution both constructed explicitly from the marginal beliefs. We showed that the freedom in the fractional parameter is useful, e.g. for finding optimal value of the parameter, $\lambda_*$, where the multipicative correction is unity. Our theory-validated experiments result in a number of interesting observation, such as strong suppression of fluctuations of $\lambda_*$ in large ensembles. We also demonstrate how the FBP approach can efficiently and more accurately solve the de-noising problem in machine learning compared to BP and TRW approaches. Finally, the combination of theoretical and empirical observations led us to hypothesize that the optimal value of $\lambda_*$, computed for a single graphical model within an ensemble, provides asymptotically exact values (in the limit of large and increasing system size) not only for the partition function but also for the marginals.

As a path forward, we envision extending this fractional approach along the following directions:

- Proving or disproving the concentration conjecture and small number of samples conjecture, made informally in Section 5.4 and Section 5.5, respectively.

- Generalizing the interpolation technique, e.g. building a scheme interpolating between TRW and Mean-Field (see e.g. Chapter 5 of Wainwright & Jordan (2008)). This will be of special interest for the case of the mixed ensembles which are generally out of reach of the fractional approach (between TRW and BP) presented in the manuscript.

- Generalizing the interpolation technique to a more general class of Graphical Models.

We also anticipate that all of these developments, presented in this manuscript and others to follow, will help to make variational GM techniques competitive with other, and admittedly more popular, methods of Machine Learning, such as Deep Learning (DL). We foresee that in the future, there will be more examples where variational GM techniques will be enhanced with automatic differentiation, e.g. in the spirit of (Lucibello et al., 2022), and also integrated into modern Deep Learning protocols, e.g. as discussed in(Garcia Satorras & Welling, 2021). This hybrid GM-DL approach is expected to be particularly beneficial and powerful in physics problems where we aim to learn reduced models with graphical structures prescribed by the underlying physics from data.

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

# A Fractional Variational Formulation: Details

## A.1 Lagrangian Formulation

Introducing Lagrangian multipliers associated with the linear constraints in Eqs. (9a,9b) we arrive at the following Lagrangian reformulation of Eq. (8)

$$\bar{F}^{(\lambda)} = \min_{\mathcal{B} \geq 0} \max_{\boldsymbol{\eta}, \boldsymbol{\psi}} L^{(\lambda)}(\mathcal{B}; \boldsymbol{\eta}, \boldsymbol{\psi}), \quad L^{(\lambda)} := F^{(\lambda)}(\mathcal{B}) + \tag{14}$$

$$\sum_{a \in \mathcal{V}; b \sim a} \sum_{x_a} \eta_{b \to a}(x_a) \left( \sum_{x_b} \mathcal{B}_{ab}(x_a, x_b) - \mathcal{B}_a(x_a) \right) + \sum_{\{a,b\} \in \mathcal{E}} \psi_{ab} \left( 1 - \sum_{x_a, x_b} \mathcal{B}_{ab}(x_a, x_b) \right),$$

where $L^{(\lambda)}(\mathcal{B}; \boldsymbol{\eta}, \boldsymbol{\psi})$ is the (extended) Lagrangian dependent on both the primary variables (beliefs, $\mathcal{B}$) and the newly introduced dual variables, $\boldsymbol{\eta} := (\eta_{b \to a}(x_a) \in \mathbb{R} | \forall a \in \mathcal{V}, \forall b \sim a, \forall x_a = \pm 1)$ and $\boldsymbol{\psi} := (\psi_a \in \mathbb{R} | \forall a \in \mathcal{V})$. The stationary point of the Lagrangian (14), assuming that it is unique, is defined by the following system of equations

$$\forall \{a, b\} \in \mathcal{E}, \ \forall x_a, x_b = \pm 1 : \quad \frac{\delta L^{(\lambda)}(\mathcal{B})}{\delta \mathcal{B}_{ab}(x_a, x_b)} = 0 \Rightarrow E_{ab}(x_a, x_b) + \tag{15}$$

$$\rho_{ab}^{(\lambda)} \left( \log \left( \mathcal{B}_{ab}^{(\lambda)}(x_a, x_b) \right) + 1 \right) - \psi_{ab}^{(\lambda)} + \eta_{b \to a}^{(\lambda)}(x_a) + \eta_{a \to b}^{(\lambda)}(x_b) = 0,$$

$$\forall a \in \mathcal{V}, \ \forall x_a = \pm 1 : \ \frac{\delta L^{(\lambda)}(\mathcal{B})}{\delta \mathcal{B}_a(x_a)} = 0 \Rightarrow \left( \sum_{b \sim a} \rho_{ab}^{(\lambda)} - 1 \right) \left( \log \mathcal{B}_a^{(\lambda)}(x_a) + 1 \right) + \sum_{b \sim a} \eta_{b \to a}^{(\lambda)}(x_a) = 0, \tag{16}$$

augmented with Eqs. (9a,9b). Eqs. (15) and Eqs. (16) result in the following expressions for the marginals in terms of the Lagrangian multipliers

$$\forall a \in \mathcal{V}, \ \forall x_a = \pm 1 : \quad \mathcal{B}_a^{(\lambda)}(x_a) \propto \exp \left( - \frac{\sum_{b \sim a} \eta_{b \to a}^{(\lambda)}(x_a)}{\sum_{b \sim a} \rho_{ab}^{(\lambda)} - 1} \right), \tag{17}$$

$$\forall \{a, b\} \in \mathcal{E}, \ \forall x_a, x_b = \pm 1 : \ \mathcal{B}_{ab}^{(\lambda)}(x_a, x_b) \propto \exp \left( - \frac{E_{ab}(x_a, x_b) + \eta_{b \to a}^{(\lambda)}(x_a) \eta_{a \to b}^{(\lambda)}(x_b)}{\rho_{ab}^{(\lambda)}} \right). \tag{18}$$

Here in Eqs. (15,16,17,18) and below, the upper index $(\lambda)$ in $\mathcal{B}^{(\lambda)}, \eta^{(\lambda)}$ and $\psi^{(\lambda)}$ variables indicates that the respective variables are optimal, i.e. argmax and argmin, over respective optimizations in Eq. (14).

### A.2 Message Passing

We may also rewrite Eqs. (17,18) in terms of the so-called message (from node to node) variables. Then the marginal beliefs are expressed via the $\mu^{(\lambda)}$-messages according to

$$\forall a \in \mathcal{V},\ \forall b \sim a:\ \mu_{b \to a}^{(\lambda)}(x_a) := \exp\left(-\frac{\eta_{b \to a}^{(\lambda)}(x_a)}{\sum\limits_{b \sim a} \rho_{ab}^{(\lambda)} - 1}\right), \tag{19}$$

$$\forall a \in \mathcal{V},\ \forall x_a = \pm 1:\ \mathcal{B}_a^{(\lambda)}(x_a) = \frac{\prod\limits_{b \sim a} \mu_{b \to a}^{(\lambda)}(x_a)}{\sum\limits_{x'_a} \prod\limits_{b \sim a} \mu_{b \to a}^{(\lambda)}(x'_a)}, \tag{20}$$

$$\forall \{a,b\} \in \mathcal{E},\ \forall x_a, x_b = \pm 1:\ \mathcal{B}_{ab}^{(\lambda)}(x_a, x_b) \tag{21}$$

$$= \frac{\exp\left(-\frac{E_{ab}(x_a,x_b)}{\rho_{ab}^{(\lambda)}}\right) \left(\mu_{b \to a}^{(\lambda)}(x_a)\right)^{\frac{\sum\limits_{c \sim a} \rho_{ac}^{(\lambda)} - 1}{\rho_{ab}^{(\lambda)}}} \left(\mu_{a \to b}^{(\lambda)}(x_b)\right)^{\frac{\sum\limits_{c \sim b} \rho_{bc}^{(\lambda)} - 1}{\rho_{ab}^{(\lambda)}}}}{\sum\limits_{x'_a, x'_b} \exp\left(-\frac{E_{ab}(x'_a,x'_b)}{\rho_{ab}^{(\lambda)}}\right) \left(\mu_{b \to a}^{(\lambda)}(x'_a)\right)^{\frac{\sum\limits_{c \sim a} \rho_{ac}^{(\lambda)} - 1}{\rho_{ab}^{(\lambda)}}} \left(\mu_{a \to b}^{(\lambda)}(x'_b)\right)^{\frac{\sum\limits_{c \sim b} \rho_{bc}^{(\lambda)} - 1}{\rho_{ab}^{(\lambda)}}}},$$

and the Fractional Belief Propagation (FBP) equations, expressing relations between pairwise and singleton marginals become:

$$\forall a \in \mathcal{V},\ \forall b \sim a,\ \forall x_a = \pm 1:\quad \mathcal{B}_a^{(\lambda)}(x_a) \propto \prod_{b \sim a} \mu_{b \to a}^{(\lambda)}(x_a) \tag{22}$$

$$\propto \sum_{x_b} \exp\left(-\frac{E_{ab}(x_a,x_b)}{\rho_{ab}^{(\lambda)}}\right) \left(\mu_{b \to a}^{(\lambda)}(x_a)\right)^{\frac{\sum\limits_{c \sim a} \rho_{ac}^{(\lambda)} - 1}{\rho_{ab}^{(\lambda)}}} \left(\mu_{a \to b}^{(\lambda)}(x_b)\right)^{\frac{\sum\limits_{c \sim b} \rho_{bc}^{(\lambda)} - 1}{\rho_{ab}^{(\lambda)}}} \propto \sum_{x_b} \mathcal{B}_{ab}^{(\lambda)}(x_a, x_b).$$

Note (on a tangent), that the $\mu^{(\lambda)}$-(message) variables introduced here are related but not equivalent to the $M^{(\lambda)}$-messages which can also be seen used in the BP-literature, see e.g. Section 4.1.3 of (Wainwright & Jordan, 2008). Specifically in the case of BP, i.e. when $\rho_{ab}^{(\lambda)} = 1$, relation between $\mu^{(\lambda)}$ and $M^{(\lambda)}$ variables is as follows, $(\mu_{b \to a}^{(\lambda)}(x_a))^{d_a - 1} = \prod_{c \sim a; c \neq b} M_{c \to a}^{(\lambda)}(x_a)$.

## B  Proof of Theorem 3.1

Let us evaluate the derivative of the fractional free energy (8) over $\lambda$ explicitly

$$\frac{d}{d\lambda} \bar{F}^{(\lambda)} = \frac{d}{d\lambda} F^{(\lambda)}\left(\mathcal{B}^{(\lambda)}\right) = \sum_{\{a,b\}} \sum_{x_a, x_b} \frac{\partial F^{(\lambda)}\left(\mathcal{B}^{(\lambda)}\right)}{\partial \mathcal{B}_{ab}^{(\lambda)}(x_a, x_b)} \frac{d\mathcal{B}_{ab}^{(\lambda)}(x_a, x_b)}{d\lambda}$$

$$+ \sum_a \sum_{x_a} \frac{\partial F^{(\lambda)}\left(\mathcal{B}^{(\lambda)}\right)}{\partial \mathcal{B}_a^{(\lambda)}(x_a)} \frac{d\mathcal{B}_a^{(\lambda)}(x_a)}{d\lambda} - \sum_{\{a,b\}} \frac{\partial H^{(\lambda)}(\mathcal{B}^{(\lambda)})}{\partial \rho_{ab}^{(\lambda)}} \frac{d\rho_{ab}^{(\lambda)}}{d\lambda}.$$

Taking into account the conditions of stationarity of the fractional free energy, tracking explicit dependencies of the fractional entropy on $\rho_{ab}^{(\lambda)}$, and thus on $\lambda$, we arrive at

$$\forall\{a,b\}: \quad \frac{\partial F^{(\lambda)}\left(\mathcal{B}^{(\lambda)}\right)}{\partial \mathcal{B}_{ab}^{(\lambda)}(x_a, x_b)} = 0; \quad \forall a: \quad \frac{\partial F^{(\lambda)}\left(\mathcal{B}^{(\lambda)}\right)}{\partial \mathcal{B}_a^{(\lambda)}(x_a)} = 0;$$

$$\frac{\partial H^{(\lambda)}(\mathcal{B}^{(\lambda)})}{\partial \rho_{ab}^{(\lambda)}} = - \sum_{x_a,x_b=\pm 1} \mathcal{B}_{ab}^{(\lambda)}(x_a, x_b) \log \mathcal{B}_{ab}^{(\lambda)}(x_a, x_b) + \sum_{x_a=\pm 1} \mathcal{B}_a^{(\lambda)}(x_a) \log \mathcal{B}_a^{(\lambda)}(x_a)$$
$$+ \sum_{x_b=\pm 1} \mathcal{B}_b^{(\lambda)}(x_b) \log \mathcal{B}_b^{(\lambda)}(x_b) = -I_{ab}^{(\lambda)},$$

where the newly introduced $I_{ab}^{(\lambda)}$ is nothing but the pairwise mutual information defined according to $\mathcal{B}^{(\lambda)}$. Notice that $I_{ab}^{(\lambda)} \geq 0$. Since, according to the theorem's assumption $\mathcal{B}^{(\lambda)}$ is differentiable in $\lambda$; $d\rho_{ab}^{(\lambda)}/d\lambda = 1 - \rho_{ab} \geq 0$, and then summarizing all of the above we derive

$$\frac{d}{d\lambda}\bar{F}^{(\lambda)} = - \sum_{\{a,b\}} (1 - \rho_{ab})I_{ab}^{(\lambda)} \leq 0, \tag{23}$$

thus concluding the proof of both continuity (the derivative is bounded) and monotonicity (the derivative is negative).

## C  Proof of Theorem 4.1

Consistently with Eq. (5), Eqs. (20,21) allow us to rewrite the joint probability distribution in terms of the optimal beliefs which solve the fractional Eqs. (22)

$$p(\boldsymbol{x}) = Z^{-1} \prod_{\{a,b\}\in\mathcal{E}} \left( \sum_{x_a,x_b} \exp\left(-\frac{E_{ab}(x_a, x_b)}{\rho_{ab}^{(\lambda)}}\right) \left(\mu_{b\to a}^{(\lambda)}(x_a)\right)^{\frac{\sum_{c\sim a}\rho_{ac}^{(\lambda)}-1}{\rho_{ab}^{(\lambda)}}} \left(\mu_{a\to b}^{(\lambda)}(x_b)\right)^{\frac{\sum_{c\sim b}\rho_{bc}^{(\lambda)}-1}{\rho_{ab}^{(\lambda)}}} \right)^{\rho_{ab}^{(\lambda)}} \times$$

$$\prod_{a\in\mathcal{V}} \left( \sum_{x_a} \prod_{b\sim a} \mu_{b\to a}^{(\lambda)}(x_a) \right)^{1-\sum_{c\sim a}\rho_{ac}^{(\lambda)}} \frac{\prod_{\{a,b\}\in\mathcal{E}} \left(\mathcal{B}_{ab}^{(\lambda)}(x_a, x_b)\right)^{\rho_{ab}^{(\lambda)}}}{\prod_{a\in\mathcal{V}} \left(\mathcal{B}_a^{(\lambda)}(x_a)\right)^{\sum_{c\sim a}\rho_{ac}^{(\lambda)}-1}}. \tag{24}$$

Normalization condition requires the sum on the right hand side of Eq. (24) to return 1, results in the desired statement, i.e. Eqs. (11,12).

## D  More Figures from Numerical Experiments

Fig. 5, mentioned in Section 5.4 shows $\lambda_*$ for a number of instances drawn from the respective ensembles of the Ising model (over planar and complete graphs). We observe that in the planar case, $\lambda_* \in [0.75, 0.95]$, while in the case of the complete graph, $\lambda_* \in [0.05, 0.15]$.

Fig. 6, mentioned in Section 5.6, shows results for the mixed case when the pair-wise interaction can vary in sign from edge to edge. In this mixed case, as seen in the presented examples, we can not guarantee that BP provides a lower bound on the partition function, and thus $\lambda_*$ may or may not be identified within the $[0, 1]$ interval.

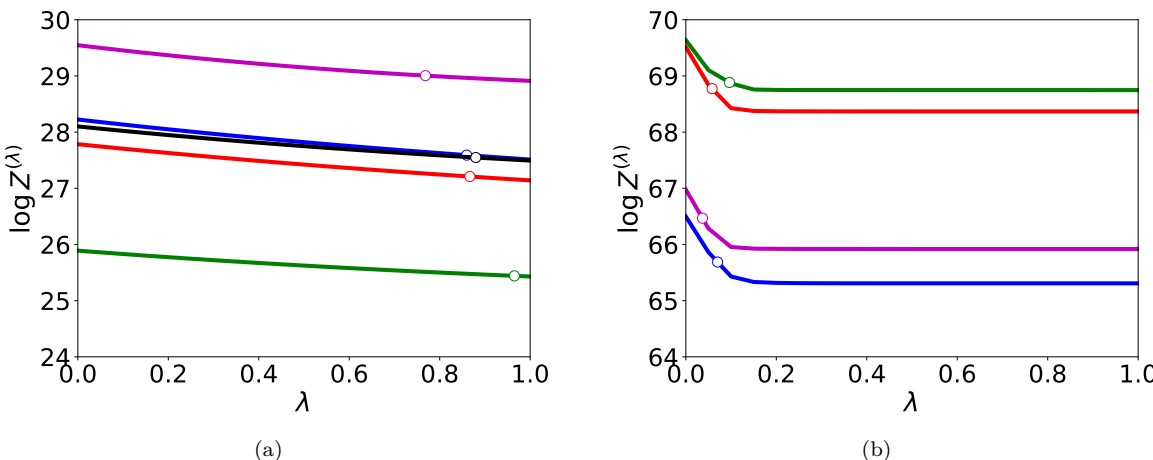

Figure 5: $F^{(\lambda)}$ vs $\lambda$ for a number of instances (shown in different colors) drawn for the Ising model ensembles over, (a) $3 \times 3$ grid, and (b) $K_9$ graph, where elements of $\boldsymbol{J}$ and $\boldsymbol{h}$ are i.i.d. from $\mathcal{U}(0,1)$. Circles mark respective exact values, $\lambda_*$.

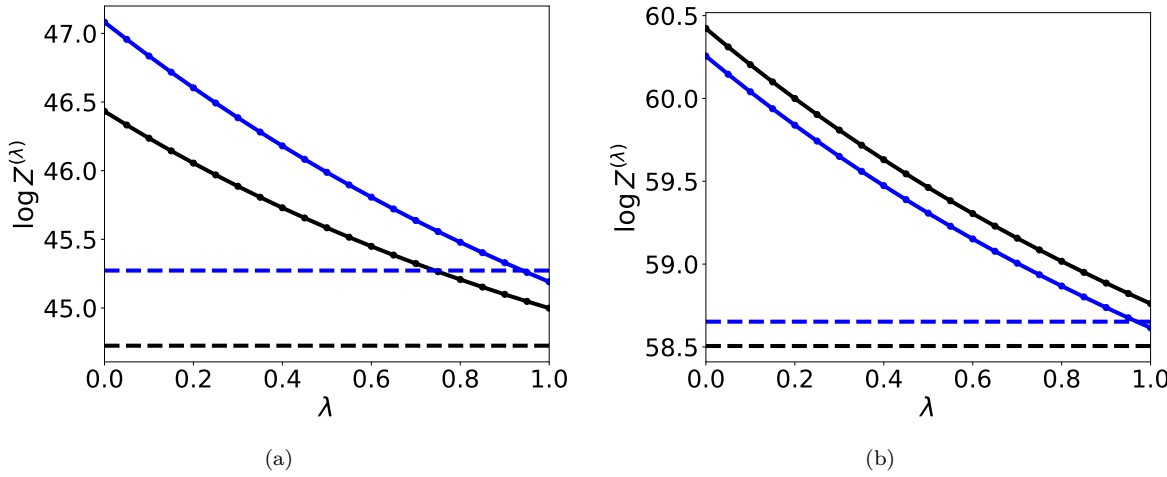

Figure 6: Two different random instance of $4 \times 4$ Isisng Model with (a) $J \sim \mathcal{U}(-1,1)$ and $h \sim \mathcal{U}(-1,1)$ (b) $J \sim \mathcal{U}(-1,1)$, $h = 0$. Dashed line shows exact value of partition functions for the corresponding curve.

