# OpenReview forum: "Exact Fractional Inference via Re-Parametrization \& Interpolation between Tree-Re-Weighted- and Belief Propagation- Algorithms"
_TMLR — Accepted by TMLR_

### Review · Reviewer_NNDJ · 2024-05-02

**Summary Of Contributions:**

This paper introduces a new message passing algorithm for computing the partition function of Ising models, understood here as graphical models on an arbitrary finite graph with pairwise and one-vertex interactions, which may be edge resp. vertex dependent. In the most general setting the edge interactions can be ferromagnetic, anti-ferromagnetic or discorded.

The algorithm arises by interpolating between two previously known message passing algorithms: Belief Propagation (BP) and Tree Re-Weighted BP (TRWBP). In the presentation of the paper both of these algorithms correspond to different ways to parameterize a class of "trial measures" to approximate the Gibbs measure. A trial measure is found using BP message passing, and then plugged into the Gibbs variational principle to estimate the partition function. The authors write the trial measure parameterization of BP and TRWBP in the unified formula (8), which depends on an interpolation parameter lambda. The values $\lambda=1$ and $\lambda=0$ correspond to the parameterization of BP and TRWBP respectively. They then consider this formula for $\lambda$ between $0$ and $1$, and note that for each $\lambda \in [0,1]$ one obtains a different trial measure parameterization, and therefore a different message passing algorithm.

In the ferromagnetic case they make the interesting observation that the true partition function $Z$ is equal to $Z^{(\lambda)}$ for some lambda in $[0,1]$, where $Z^{(\lambda)}$ denotes the estimate of the partition function with the optimal parameters (=optimal messages) for the corresponding trial measure parameterization. [Lemma 3.3]

Furthermore, they obtain a kind of factorization formula for $Z$ that holds for any $\lambda$. It says that for all lambda, $Z = Z^{(\lambda)} \times \mathcal{Z}^{(\lambda)}$, where $\mathcal{Z}^{(\lambda)}$ is a quantity that can be computed from the optimal parameters/messages for that $\lambda$. [Theorem 4.1].

Based on this they formulate their new algorithm for estimating the parition function $Z$ which they name the Fractional Message Passing Algorithm (FBP): [Algorithm 1]

For a range of $\lambda \in [0,1]$:
- Use message passing to estimate the optimal parameters/messages for $\lambda$.
- Compute estimates of $Z^{(\lambda)}$ and $\mathcal{Z}^{(\lambda)}$ using these.

At the end, for $\lambda$ such that the estimate of $\mathcal{Z}^{(\lambda)}$ is closest to one, output the estimate of $Z^{(\lambda)}$.

The success of the algorithm is contingent on the convergence of the message passing algorithm for each $\lambda$.

As far as I can tell, for the ferromagnetic case, in an idealized setting where an oracle gives exact optimal parameters/messages for each $\lambda$, and using the oracle is used for every value of $\lambda$ in $[0,1]$, this algorithm is guaranteed to return the true value of the partition function $Z$.

Beyond the ferromagnetic case, there is no guaranteed that $Z = Z^{(\lambda)}$ for some $\lambda \in [0,1]$, so the algorithm may fail, but it may also plausibly produce good estimates in some cases.

The authors present numerical experiments for different "toy" instances of the Ising model. These "toy" instances seem to be more the spirit of theoretical physics than machine learning.

**Audience:**

Yes

**Broader Impact Concerns:**

None.

**Claims And Evidence:**

Yes

**Requested Changes:**

Please improve presentation.
* There are a few places in the text where I cannot figure out exactly
what is meant. Please improve the corresponding formulations.
   - In step 2 of the for loop of Algorithm 1 it says: use TRWBP to find
$Z^{(\lambda)},\ldots$'. Is this a separate use of TRW, distinct
from formulating eq. (8)? So that there is a kind of ``TRW on top
of TRW'' in the algorithm? Also, why not use simply BP in this step?
        - **[Edit]** In fact, to make the algorithm fully unambiguous also for readers who are not so used to thinking about message passing algorithms, it would be helpful if step 2 is fleshed out and written more explicitly. If the authors don't want to put such detail in algorithm 1, they could write a detailed "subalgorithm" in the appendix that the pseudocode in algorithm 1 can "call".
   - In Section 6 in the second bullet point it says ``building a scheme
interpolating between TRW and Mean-Field''. What is meant by mean
field here?
   - The abbreviation FBP is used on p6 before it has been introduced.

Please improve organization.
* Algorithm 1 should come before Section 5 on "numerical experiments". After all, it is needed to fully specify what "FBP" is.

Please improve readability of the graphs.
* One has to really look closely to see the difference between $\log Z^{(\lambda)}$
and $\log\mathcal{Z}^{(\lambda)}$ on two sides of e.g. Figure 1 (a)
(b). Before that the reader is quite puzzled by what they are supposed
to be showing. Please find a solution for this!
 * In Figure 2 I don't understand what is mean by "hollow markers''
and "open symbols''. I see the hollow small shapes on the curves,
but if they are "hollow markers'', what are the "open symbols''?
Furthermore these hollow/open symbols are much too small and difficult
to make out. Here again the reader will likely spend some time puzzling
over how exactly the "exact values'' are "values of lambda''
are supposed to be expressed.


Please improve the English of the manuscript.


* Several times terms that seem unidiomatic are used.
   - For instance I've never seen "inference efforts'' used. "Computational
complexity of inference'' or similar would make more sense to me.
   - Is it really meaningful to use the word "homotopy'' here? Maybe
there is some use of this word in a similar setting that I am not
aware of, but if not it seems like a stretch.

* Grammar: Lots of grammar mistakes.
  - In particular, often a definitive article is missing. For instance
on p. 4 it says "Moreover, fractional approach developed in this
manuscript'' instead of "Moreover, *the* fractional approach
developed in this manuscript''.
   - Sometimes there is a superfluous definitive article, like in p. 10
where it says "We envision seeing in the future more examples where
*the* variational GM techniques will be reinforced with *the*
automatic differentiation, e.g. in the spirit of (Lucibello et al.,
2022), and also integrated into modern Deep Learning protocols..''.
Both of these "the'' should be removed.
 - Ideally the manuscript should be proofread by a native speaker. If this is not possible then then please use the grammar checking features of e.g. Word to find the errors.

Please provide a comparison to other algorithms.
* Is there some benchmark problem in this area that could be used to
compare the performance of this algorithm to other algorithms? If
so please provide an experimental comparison.


Please make an application to machine learning more plausible.
* Please come up with some simple experiment that you can add that illustrates
how the algorithm might be used in a machine learning setting. As
this is a fairly theoretical work, a "toy'' example would be fine.

**Strengths And Weaknesses:**

Starting with the strengths, the idea behind the new algorithm is pleasantly elegant. The proofs seem mostly similarly elegant. For the ferromagnetic case Theorem 4.1 + Lemma 3.3 provide quite a nice theoretical guarantee (which however does not address convergence of the message passing algorithm to the optimum for each $\lambda$, but rather assumes it). As far as I can tell the idea is novel (though I am not an expert in the area of applying message passing to graphical models - for “calibration” I might mention that I was quite familiar with the BP algorithm before reading this work, but not had not heard of the TRWBP algorithm). From a theoretical stand-point I certainly find the work interesting.

A major weakness is the quality of the write-up and presentation. There are many language errors, and suboptimally presented graphs.

Another weakness is that the performance of the algorithm is not really clearly compared to other algorithms.

A further weakness is that practical applicability for actual machine learning is not made plausible. The experiments are all “toy” instances of the Ising model with no clear machine learning application. My positive answer to the "audience" question below is made with low confidence.

---

> ### Author Response · Authors · 2024-06-09
> **Responces to Reviewer NNDJ**
>
> # Please improve presentation.
>
> **In step 2 of the for loop of Algorithm 1 it says: use TRWBP to find $Z^{(\lambda)},\ldots$'. Is this a separate use of TRW, distinct from formulating eq. (8)? So that there is a kind of ``TRW on top of TRW'' in the algorithm? Also, why not use simply BP in this step?**
>
> **In fact, to make the algorithm fully unambiguous also for readers who are not so used to thinking about message passing algorithms, it would be helpful if step 2 is fleshed out and written more explicitly. If the authors don't want to put such detail in algorithm 1, they could write a detailed "subalgorithm" in the appendix that the pseudocode in algorithm 1 can "call"**.
>
> We are grateful to the reviewer for helping us identify a misprint. It was meant to be a solution of the FBP problem, not of its BP or TRW versions. We have corrected the misprint.
>
>
> We believe that correcting the aforementioned misprint and providing explicit references to the relevant formula will mitigate any potential confusion for readers unfamiliar with belief propagation.
>
> **In Section 6 in the second bullet point it says ``building a scheme interpolating between TRW and Mean-Field''. What is meant by mean field here?**
>
> Mean-field is an alternative approximation widely used in the graphical model literature. We have added reference to Wainwright and Jordan book discussing mean-field approximation in details in chapter 5.
>
> **The abbreviation FBP is used on p6 before it has been introduced.**
> This was an omission, which has now been corrected.
>
> # Please improve organization
>
> **Algorithm 1 should come before Section 5 on "numerical experiments". After all, it is needed to fully specify what "FBP" is.**
>
> We have followed the reviewer's suggestion and moved the algorithm to a position in the text that precedes the numerical experiments.
>
> # Please improve readability of the graphs.
>
> **One has to really look closely to see the difference between $\log Z^{(\lambda)}$ and $\log\mathcal{Z}^{(\lambda)}$ on two sides of e.g. Figure 1 (a) (b). Before that the reader is quite puzzled by what they are supposed to be showing. Please find a solution for this!**
>
> We agree with the reviewer and thus substituted ${\cal Z}$ by ${\tilde Z}$ through out the text.
>
> **In Figure 2 I don't understand what is mean by "hollow markers'' and "open symbols''. I see the hollow small shapes on the curves, but if they are "hollow markers'', what are the "open symbols''? Furthermore these hollow/open symbols are much too small and difficult to make out. Here again the reader will likely spend some time puzzling over how exactly the "exact values'' are "values of lambda'' are supposed to be expressed.**
>
> We thank the reviewer for identifying the ambiguity. We have revised the text, retaining the term 'open symbols' and removing 'hollow markers.' Additionally, we have enhanced the legibility of the figure.
>
> # Please improve the English of the manuscript.
>
> In response to the reviewer's feedback, we are replacing the term ``homotopy" with "interpolation".
> We are grateful to the reviewer for the English tips. We have proofread the text and corrected the grammar accordingly.
>
> # Please provide a comparison to other algorithms
> **Is there some benchmark problem in this area that could be used to compare the performance of this algorithm to other algorithms? If so please provide an experimental comparison.**
>
> We believe that the material presented in Section 5.3 addresses the reviewer's concern. In this section, we compare the performance of our Algorithm 1, which provides an approximation to the exact partition function, with its exact value on examples of Ising models over relatively small graphs that can be evaluated explicitly by a brute-force algorithm. We also show that, in these cases, our Algorithm 1 compares favorably to BP and TRW-BP as an approximation to  $Z$.
>
> # Please make an application to machine learning more plausible.
> **Please come up with some simple experiment that you can add that illustrates how the algorithm might be used in a machine learning setting. As this is a fairly theoretical work, a "toy'' example would be fine.**
>
> We are grateful to the reviewer for this suggestion, which inspired us to apply the FBP approach to a classical machine learning problem: image de-noising. According to our newly conducted experiments, the basic FBP algorithm outperforms BP and TRW-BP in this task. We have added Section 5.7 to describe these new results.

---

### Review · Reviewer_gKSN · 2024-05-26

**Summary Of Contributions:**

The paper proposes a new inference algorithm for Ising models that interpolates between two classical inference algorithms: the tree-reweighted algorithm and the belief propagation algorithm. The inference algorithm is concerned with calculating the partition function of the Ising model. Theoretical justification and empirical evaluation are provided to validate the proposed method.

**Audience:**

Yes

**Broader Impact Concerns:**

no ethical implication.s

**Claims And Evidence:**

Yes

**Requested Changes:**

* the title of section 1.2 suggests that it is concerned with the major contribution of the paper. However, a lot of background-related descriptions are also included. This makes it hard to parse the key contribution of the paper.
* The last sentence of section 4 can be broken down into multiple sentences.
* The graphs in the experiments can use some legends. The curves in figure 2 are getting too close to each other.

**Strengths And Weaknesses:**

strength:
* the paper deals with an interesting problem concerning the inference of Ising models.
* the mathematical results obtained in the paper seem to provide a nice complement and extension to existing results.
* discussion of related work is comprehensive.

weakness:
* Although understandable, experiments are conducted on synthetic data only.
* presentation can be improved.

---

> ### Author Response · Authors · 2024-06-08
> **Responces to Reviewer gKSN**
>
> Thanks a lot for your comments. We enhanced the presentation of the paper and add a section about application of our method in machine learning. The changes are shown in  blue.
>
> **the title of section 1.2 suggests that it is concerned with the major contribution of the paper. However, a lot of background-related descriptions are also included. This makes it hard to parse the key contribution of the paper**
>
> The amount of background information introduced in this part of the manuscript is relatively
> small and, in our opinion, does not warrant splitting it from the description of the manuscript’s contributions. We also believe that the background material is not disruptive; on the contrary, it helps the reader to contrast and better appreciate the contributions made in the manuscript.
>
> **The last sentence of section 4 can be broken down into multiple sentences**
>
> We follow suggestion of the reviewer and broke down the sentence into multiple sentences.
>
>
> **The graphs in the experiments can use some legends. The curves in figure 2 are getting too close to each other.**
>
> All figures have been modified to improve legibility

---

### Review · Reviewer_Nu2b · 2024-07-02

**Summary Of Contributions:**

This paper proposes a one-parameter extension,
with parameter $\lambda\in[0,1]$, of the Bethe
and the tree reweighted (TRW) approximations
of the free energy,
containing the latter two as special cases ($\lambda=1$ and 0, respectively).
The extension of the Bethe free energy,
which is called the fractional free energy
and denoted by $\bar{F}^{(\lambda)}(\mathcal{B})$
with a test pseudomarginal $\mathcal{B}$,
is formulated in equations (11)-(14),
and the corresponding extension of the belief propagation (BP) algorithm,
which is called the fractional belief propagation,
is derived as equation (26) in Appendix A.2,
via the Lagrangian formulation in Appendix A.1.
As theoretical contributions,
this paper claims to have proven continuity and monotonicity (Theorem 3.1),
as well as concavity (Theorem 3.2),
of the (optimized) fractional free energy
$\bar{F}^{(\lambda)}:=\min_{\mathcal{B}}\bar{F}^{(\lambda)}(\mathcal{B})$
as a function of the parameter $\lambda$.
It also claims to have shown that the reparametrization properties
of the Bethe approximation (namely, the fact that the original distribution is
represented, or ``reparametrized,'' in terms of any fixed point
of the Bethe free energy, see Proposition 4.3 in Wainwright and Jordan (2007))
also holds for their extension (Theorem 4.1).

As a possible application of their extension,
the authors consider the problem of estimating the partition function $Z$
of Ising models with ``attractive'' two-spin interactions (i.e.,
all the interactions $J_{ab}$ are nonnegative),
on the basis of the known observations, summarized in Lemma 3.3,
that the TRW estimate $Z^{(0)}$ and the BP estimate $Z^{(1)}$
satisfy $Z^{(0)}\ge Z$ and $Z^{(1)}\le Z$, respectively,
the latter of which has been proven to hold in Ruozzi (2012)
only for attractive Ising models.
Although the proposed algorithm for approximately evaluating $Z$ (Algorithm 1),
called the fractional message passing algorithm,
is not described clearly enough,
results of numerical experiments seem to suggest validity
of the theoretical development in this paper.

**Audience:**

Yes

**Broader Impact Concerns:**

I do not see any concerns on the ethical implications of this work, as it is of theoretical nature.

**Claims And Evidence:**

No

**Requested Changes:**

All the comments listed in "Strengths and weaknesses" above are critical to securing my recommendation for acceptance.

### [C1] Novelty
The proposed formulation of the fractional free energy
(equations (11)-(14)) is not entirely new:
The same form of an approximate free energy has been studied in
Wainwright, Jaakkola, and Willsky (2005) (see equation (25))
and Wainwright and Jordan (2008) (see, e.g., Theorem 7.2)
under the TRW rules,
as well as in Wiegerinck and Heskes (2003) (see equation (14))
where $\{\rho_{ab}\}$ are no longer assumed to follow the TRW rules.
The proposed formulation in this paper
is therefore regarded as a one-parameter subclass of
those general formulations, the former being defined by linearly interpolating
$\{\rho_{ab}\}$ satisfying the TRW rules and $\rho_{ab}=1$ which
corresponds to the Bethe free energy.
The associated belief propagation formulae and/or
fixed-point equations have been derived there as well.
These facts, as well as the relationship between
these arguments and those in this paper,
have not been stated clearly in this paper.

### [C2] Further possible extension
In view of the argument given in [C1],
one might be able to consider the following further extension.
Let $\rho:=\\\{\rho_{ab}\in[0,1]\\\}$ be completely tunable parameters,
as in Wiegerinck and Heskes (2003),
and define $\bar{F}^{(\rho)}$ and $F^{(\rho)}(\mathcal{B})$
analogously as in equations (11)-(14).
This setting corresponds to the TRW if $\rho$ satisfies
the TRW rules, to the BP if $\rho_{ab}=1$ for all $(a,b)\in\mathcal{E}$,
and to what is called the naive mean field if $\rho=\mathbf{0}$.
This extension should be feasible beyond the Ising models.
Furthermore, the following extended theorems might be expected to hold,
if the arguments in this paper be valid:

**Theorem R3.1:**
$\partial\bar{F}^{(\rho)}/\partial\rho_{ab}=-I_{ab}^{(\rho)}\le 0$,
provided that the required differentiability is satisfied
(see, however, [C3] below).

**Theorem R3.2:**
$\bar{F}^{(\rho)}$ is concave in $\rho$
(see also [C8]).

**Theorem R4.1:**
The reparametrization properties hold
for any fixed point of the fractional belief propagation algorithm.

Theorems 3.1, 3.2, and 4.1 will then be obtained
as corollaries of these theorems,
by restricting $\rho$ to be on a line segment connecting
a TRW weight (i.e., $\exists\mathcal{T};\rho_{ab}=\sum_{T\in\mathcal{T},(a,b)\in T}\rho_T,\sum_{T\in\mathcal{T}}\rho_T=1$)
and the BP weight (i.e., $\rho_{ab}=1$ for all $(a,b)\in\mathcal{E}$).

### [C3] Validity of Theorem 3.1 and Lemma 3.3
One may alternatively raise a question about the validity
of Theorem 3.1 in this paper.
First of all, one can regard that Theorem 3.1
is a restatement of Lemma 2 in Wainwright, Jaakkola, and Willsky (2005)
(note that the proof of Lemma 2 does not rely directly on
the assumption that $\rho$ satisfies the TRW rules):
One can then find in the latter that the existence and uniqueness
of $\mathcal{B}^{(\lambda)}$ play an important role
in guaranteeing the required differentiability,
and that the required existence and uniqueness are ensured by the tree subgraph
assumption.
If one brings the argument in the latter to that in this paper, however,
one may question the required differentiability
since it is known that the uniqueness of $\mathcal{B}^{(\lambda)}$
does not hold in general (consider the BP case $\lambda=1$ as an example),
which may cause violation of the differentiability
of $\mathcal{B}^{(\lambda)}$ with respect to $\lambda$
at some values of $\lambda$.
One can consequently raise the following criticism:
How can one guarantee the differentiability
required for the manipulation done in Appendix B to prove Theorem 3.1
to be valid, without the existence and uniqueness assumptions on $\mathcal{B}^{(\lambda)}$?
A more detailed proof which takes this point into account
would be appreciated.
As the proof of Lemma 3.3 relies on the intermediate value theorem,
which requires continuity of $Z^{(\lambda)}$ on $\lambda\in[0,1]$,
the above point should also affect the validity of Lemma 3.3.

### [C4] Fractional message passing algorithm
I do not understand several aspects of
the proposed fractional message passing algorithm (Algorithm 1):
- Line 2 in the For block:
  The authors write "Use TRW-BP", but what is TRW-BP?
  Although it is not explained explicitly in this paper at all,
  my guess is that it is nothing other than the fractional belief propagation
  algorithm described in equation (26), but then why do the authors
  use the different names?
- Line 1 in the End block:
  The authors write "Find $\lambda_*$ where $\mathcal{Z}^{(\lambda)}=1$",
  but how? According to Algorithm 1, one estimates the values of
  $\mathcal{Z}^{(\lambda)}$ via random sampling
  for 21 equispaced values of $\lambda$ from $\lambda=0.01$ to $\lambda=1$.
  Among the resulting 21 estimates $\{Z^{(\lambda=0.01)},\ldots,Z^{(\lambda=1)}\}$,
  it is quite unlikely that there exists a value of $\lambda$ with
  which the value of $\mathcal{Z}^{(\lambda)}$ is estimated
  to be exactly 1.
  One should also take care of the statistical fluctuations
  of these estimates arising from the random sampling.

### [C5] Possible alternatives
Since the proposed fractional message passing algorithm
involves random sampling in evaluating $\mathcal{Z}^{(\lambda)}$,
one might think about sevaral alternative approaches
which make use of similar random sampling schemes.
For example:
- **Direct sampling:**
  Let $q(x)$ be a tractable probability distribution
  on $\\\{-1,1\\\}^N$, and approximate $Z$ as
  \begin{equation}
    Z=\sum_{x\in\{-1,1\}^N}q(x)\frac{\exp(-E(x))}{q(x)}
      \approx\frac{1}{M}
      \sum_{i=1}^M\frac{\exp(-E(x_i))}{q(x_i)},
  \end{equation}
  where $\\\{x_1,x_2,\ldots\\\}$ is a sequence of random samples
  drawn from the distribution $q$.
  If $q$ is simple enough, then one can use iid sampling.
  One may alternatively use Markov-Chain Monte-Carlo (MCMC) sampling
  when $q$ is not simple but has a structure suitable for MCMC sampling.
  The computational complexity of the above scheme
  is at most the same as that of evaluating $\mathcal{Z}^{(\lambda)}$
  in the proposed algorithm.
- **TRW+Sampling:**
  Perform TRW to obtain $\mathcal{B}^{(0)}$ and $Z^{(0)}$,
  and then evaluate $\mathcal{Z}^{(0)}$
  via the same random sampling procedure as that of the proposed algorithm.
  Return $Z^{(0)}\mathcal{Z}^{(0)}$ as an estimate of $Z$.
  An obvious advantage of this approach is
  that the TRW is guaranteed to converge in finite time
  and yields a unique solution.
- **BP+Sampling:**
  Perform BP to obtain $\mathcal{B}^{(1)}$ and $Z^{(1)}$,
  and then evaluate $\mathcal{Z}^{(1)}$
  via the same random sampling procedure as that of the proposed algorithm.
  Return $Z^{(1)}\mathcal{Z}^{(1)}$ as an estimate of $Z$.
- **FBP+Sampling:**
  Choose a value of $\lambda\in[0,1]$ according to
  some criterion.
  Perform fractional belief propagation with the chosen value of $\lambda$
  to obtain $\mathcal{B}^{(\lambda)}$ and $Z^{(\lambda)}$,
  and then evaluate $\mathcal{Z}^{(\lambda)}$
  via the same random sampling procedure as that of the proposed algorithm.
  Return $Z^{(\lambda)}\mathcal{Z}^{(\lambda)}$ as an estimate of $Z$.
  This reduces to TRW+Sampling and BP+Sampling
  if one lets $\lambda=0$ and 1, respectively.

An apparent disadvantage of the proposed fractional message passing
algorithm (Algorithm 1) compared with the above competitors
is that the proposed algorithm requires several runs
of fractional belief propagation with several different values of $\lambda$,
which makes the computational complexity larger.
In order for the proposed algorithm to be significant,
one would have to demonstrate its superiority
compared with competitors such as those listed above.
It would be nice if the proposed algorithm
gives good results compared with those competitors,
with a similar amount of computational resources.

I am also concerned about absence of results which directly show
how good the estimates of the partition function $Z$
with the proposed approach were.

### [C6] Equation (5)
Equation (5) seems incorrect.
Consider the KL divergence $D(\mathcal{B}\|p(x|J,h))$
of a test distribution $\mathcal{B}(x)$
from the true distribution $p(x|J,h)$:
\begin{equation}
  D(\mathcal{B}\|p(x|J,h))
  =\sum_{x}\mathcal{B}(x)
  \log\frac{\mathcal{B}(x)}{p(x|J,h)}
  =\sum_{x}\mathcal{B}(x)\log\mathcal{B}(x)
  +\sum_{x}\mathcal{B}(x)E(x)+\log Z,
\end{equation}
where $Z=Z(J,h)$ is the partition function
of the true distribution as defined in equation (2).
One then has
\begin{equation}
  D(\mathcal{B}\|p(x|J,h))-\log Z
  =\sum_{x}(E(x)\mathcal{B}(x)
  +\mathcal{B}(x)\log\mathcal{B}(x)),
\end{equation}
and thus, noting the nonnegativity of the KL divergence,
\begin{equation}
  F:=-\log Z
  =\sum_{x}(E(x)\mathcal{B}(x)
  +\mathcal{B}(x)\log\mathcal{B}(x))
  -D(\mathcal{B}\|p(x|J,h))
  =\min_{\mathcal{B}(x)}
  \sum_{x}(E(x)\mathcal{B}(x)
  \mathrel{\color{red}+}\mathcal{B}(x)\log\mathcal{B}(x)),
\end{equation}
where the minimum with respect to $\mathcal{B}(x)$
is achieved at $\mathcal{B}(x)=p(x|J,h)$.
The sign just before the term $\mathcal{B}(x)\log\mathcal{B}(x)$
is not the minus $-$ as in equation (5)
but the plus $+$ as shown above.

The authors also wrote just above equation (5) that
they consider the Kullback-Leibler distance between
$\exp(-E(x;J,h))$ and $\mathcal{B}(x)$.
Although $\mathcal{B}(x)$ is a proper probability distribution,
$\exp(-E(x;J,h))$ is not in general,
so that one cannot consider the Kullback-Leibler divergence
with it.

### [C7] Derivation of fractional free energy
Substituting equation (8) into equation (5) does not yield equations (11)-(14).
It is obvious from the fact that
$\mathcal{B}(x)$ as defined in equation (8)
is not in general a proper probability distribution,
as mentioned by the authors themselves in page 5, lines 3--4.
Also, the argument $\mathcal{B}$ of $F^{(\lambda)}$ defined in equation (11)
is not the function $\mathcal{B}(x)$ defined in equation (8)
but the collection of single- and two-body marginals $\\\{ℬ_a(x_a)\\\}$
and $\\\{ℬ_{ab}(x_a,x_b)\\\}$.
It is therefore appropriate to define $\mathcal{B}$ not as in equation (8)
but rather as a pseudomarginal
$ℬ=\\\{ℬ_a(x_a),ℬ_{ab}(x_a,x_b)\\\}$
as discussed by, e.g., Wainwright's papers.
At least one has to distinguish the test distribution $\mathcal{B}(x)$
in Section 2.2, which is a proper probability distribution,
and the pseudomarginal $\mathcal{B}$ in Section 2.3.
(Note: I occasionally used ℬ in place of $\mathcal{B}$ in the above, in order to avoid MarkDown formatting problems.)

There is notational inconsistency:
- Later the authors
use the expression $\mathcal{B}^{(\lambda)}$ to denote
the pseudomarginal optimizing $\bar{F}^{(\lambda)}(\mathcal{B})$,
as mentioned in page 13, lines 19--20,
that is, $\mathcal{B}^{(\lambda)}:=\min\bar{F}^{(\lambda)}(\mathcal{B})$,
whereas in equation (8) they use the same expression to denote
a generic pseudomarginal,
irrespectively of being optimized or not.
- An edge is represented in two different ways: $(a,b)$ and $\\\{a,b\\\}$.
  The same notation should be used throughout the paper.

### [C8] Theorem 3.2
Since the proof is straightforward, I think it better
to write it down much more concretely, as:
> As $\rho_{ab}^{(\lambda)}$ is linear in $\lambda$,
from Eq. (13) $H^{(\lambda)}(\mathcal{B})$ is also linear in $\lambda$,
so that one has, for any $\lambda,\lambda_0,\lambda_1$ with
$\lambda_0<\lambda_1$,
\begin{equation}
  H^{(\lambda)}(\mathcal{B})
  =\frac{\lambda_1-\lambda}{\lambda_1-\lambda_0}H^{(\lambda_0)}(\mathcal{B})
    +\frac{\lambda-\lambda_0}{\lambda_1-\lambda_0}H^{(\lambda_1)}(\mathcal{B}),
\end{equation}
and therefore
\begin{equation}
  F^{(\lambda)}(\mathcal{B})
  =\frac{\lambda_1-\lambda}{\lambda_1-\lambda_0}F^{(\lambda_0)}(\mathcal{B})
    +\frac{\lambda-\lambda_0}{\lambda_1-\lambda_0}F^{(\lambda_1)}(\mathcal{B}).
\end{equation}
One consequently has
\begin{equation}
  \bar{F}^{(\lambda)}
  =F^{(\lambda)}(\mathcal{B}^{(\lambda)})
  =\frac{\lambda_1-\lambda}{\lambda_1-\lambda_0}
  F^{(\lambda_0)}(\mathcal{B}^{(\lambda)})
  +\frac{\lambda-\lambda_0}{\lambda_1-\lambda_0}
  F^{(\lambda_1)}(\mathcal{B}^{(\lambda)})
  \ge\frac{\lambda_1-\lambda}{\lambda_1-\lambda_0}
  F^{(\lambda_0)}(\mathcal{B}^{(\lambda_0)})
  +\frac{\lambda-\lambda_0}{\lambda_1-\lambda_0}
  F^{(\lambda_1)}(\mathcal{B}^{(\lambda_1)})
  =\frac{\lambda_1-\lambda}{\lambda_1-\lambda_0}
  \bar{F}^{(\lambda_0)}
  +\frac{\lambda-\lambda_0}{\lambda_1-\lambda_0}
  \bar{F}^{(\lambda_1)},
\end{equation}
where at the inequality sign we used the fact
$F^{(\lambda)}(\mathcal{B}^{(\lambda)})
=\min_{\mathcal{B}}F^{(\lambda)}(\mathcal{B})\le F^{(\lambda)}(\mathcal{B}^{(\lambda')})$.
It proves the concavity of $\bar{F}^{(\lambda)}$ in $\lambda$.

### [C9] Lemma 3.3
How the attractiveness condition is used is not clear.
I guess that the condition is used in Ruozzi (2012) to
prove $Z^{(1)}\le Z$, which should be explicitly stated.

### [C10] Theorem 4.1
It should be mentioned that Theorem 4.1 is an extension
of the known result of the reparametrization properties
of Bethe approximation (e.g. Proposition 4.3 of Wainwright and Jordan, 2007).

I think that equation (28) in Appendix C containing the proof
of Theorem 4.1 has the following error: The factor
\begin{equation}
  \prod_{a\in\mathcal{V}}\left(\sum_{x_a}\prod_{b\sim a}\mu_{b\to a}^{(\lambda)}(x_a)
  \right)^{\sum_{c\sim a}\rho_{ac}^{(\lambda)}-1}
\end{equation}
appearing on the right-hand side should be
\begin{equation}
  \prod_{a\in\mathcal{V}}\left(\sum_{x_a}\prod_{b\sim a}\mu_{b\to a}^{(\lambda)}(x_a)
  \right)^{\color{red}1-\sum_{c\sim a}\rho_{ac}^{(\lambda)}}.
\end{equation}

### [C11] Lemma 5.1
I think that the proof of Lemma 5.1 in Appendix D is
severely flawed in several respects.
As demonstrated by the example shown in Figure 3,
the authors first consider the $N$-node complete graph $K_N$,
and initialize $\mathcal{T}$ as the set of all linear spanning trees
of $K_N$.
Since any permutation $(\sigma(1),\sigma(2),\ldots,\sigma(N))$
of $(1,2,\ldots,N)$ corresponds to a linear spanning tree of $K_N$,
and since each linear spanning tree of $K_N$ corresponds to
two such permutations $(\sigma(1),\sigma(2),\ldots,\sigma(N))$
and $(\sigma(N),\sigma(N-1),\ldots,\sigma(1))$,
the total number $|\mathcal{T}|$ of the linear spanning trees
of $K_N$ is $N!/2$.
For $N=4$, it is equal to 12 (see a separate [figure](https://figshare.com/s/ae3c257ed285496a5946),
which shows all the linear spanning trees of $K_4$: the left six drawn in black are in Figure 3 (a),
whereas the right six drawn in red are absent).
On the other hand, in Figure 3 (a) only six of them are shown,
so that it would be inappropriate as an illustration of Algorithm 2.
Furthermore, the edge counts in those six spanning trees
are not uniform across the six edges of $K_4$:
Although two edges have edge counts of 3,
two others have 2, and the remaining two have 4.
It is inconsistent with the description in the figure caption,
as they do not yield $\rho$ satisfying the edge-uniform rule.
If instead one considers all the 12 linear spanning trees
shown in the figure,
then the edge counts are 6 for all the six edges of $K_4$.

Also, the figure caption itself contains several errors:
$|V|$ should read $|{\color{red}\mathcal{V}}|$.
The formula $\rho_{ab}=|\mathcal{V}|-1/|\mathcal{E}|$ should read
$\rho_{ab}={\color{red}(}|\mathcal{V}|-1{\color{red})}/|\mathcal{E}|$.
(b) $|V|=5$ should be $|{\color{red}\mathcal{V}}|={\color{red}4}$.

In line 2 of Algorithm 2,
the authors let $\rho_T=1/|\mathcal{E}|$.
This assertion is in general invalid, as it may violate the TRW rules:
\begin{equation}
  \sum_{T\in\mathcal{T}}\rho_T
  =\sum_{T\in\mathcal{T}}\frac{1}{|\mathcal{E}|}
  =\frac{|\mathcal{T}|}{|\mathcal{E}|}
  \stackrel{?}{=}1.
\end{equation}
Take $\mathcal{G}=K_{|\mathcal{V}|}$ as an example.
In view of the argument shown above, one has $|\mathcal{T}|=|\mathcal{V}|!/2$,
whereas the total number of edges in $K_{|\mathcal{V}|}$ is given by
$|\mathcal{E}|=|\mathcal{V}|(|\mathcal{V}|-1)/2$.
They are equal only when $|\mathcal{V}|=2$ or $|\mathcal{V}|=3$.
In general, one should let $\rho_T={\color{red}1/|\mathcal{T}|}$.

I think that the most serious problem with the proof of Lemma 5.1
is that it is not clear at all whether the procedure described
in Algorithm 2 is always feasible: For example, in line 6,
one has to "modify $\mathcal{T}$ by adding an extra edge to each spanning tree of $\mathcal{T}$'' in such a way that one ``guarantees that each edge enters
exactly $N-1$ resulting spanning trees."
There would be several possibilities of adding an edge,
some of which might result in the same spanning tree already in $\mathcal{T}$.
I do not understand either why the authors can guarantee
"that each edge enters exactly $N-1$ resulting spanning trees,"
as they did not provide any concrete proof for that.

I even think that Algorithm 2 *cannot* guarantee existence
of a subset of spanning trees with the desired properties,
as opposed to the authors' claim.
Consider a graph $\mathcal{G}=(\mathcal{V},\mathcal{E})$,
where $\mathcal{E}$ is equal to the edge set of
the complete graph $K_{|\mathcal{V}|}$ *minus*
the single edge $(N-1,N)$, where $N=|\mathcal{V}|$.
Recalling that the number of linear spanning trees of $K_N$ is $N!/2$,
after removing from $\mathcal{T}$ one linear spanning tree which becomes
disconnected by the edge removal, and then adding an extra edge
to each of the remaining linear spanning trees which have become
disconnected by the edge removal,
the total number of spanning trees in $\mathcal{T}$
should be $N!/2-1$, each with $(N-1)$ edges.
It implies that now $\mathcal{T}$ contains
$(N-1)(N!/2-1)$ edges in total.
In order to achieve the edge-uniform rule
(i.e., $\rho_{ab}$ are independent of edge $(a,b)$)
with equal-weight trees,
these edges should be distributed equally to
the $|\mathcal{E}|=N(N-1)/2-1$ edges of $\mathcal{G}$,
resulting in the average edge count of
\begin{equation}
  \frac{(N-1)(N!/2-1)}{N(N-1)/2-1}
  =\frac{(N-1)(N!-2)}{(N+1)(N-2)}\quad (*)
\end{equation}
per edge.
In order for Lemma 5.1 to be valid,
this quantity should be an integer.
As shown in the following table, however,
this quantity can be non-integer. (I numerically confirmed that
this quantity is non-integer for all $N\in\{4,5,\ldots,10000\}$.)
This fact demonstrates that Algorithm 2 does not work as expected,
therefore disproving Lemma 5.1.

|$N$|$3$|$4$|$5$|$6$|$7$|
|----|----|----|----|----|----|
|average edge count (*)|2|$\frac{33}{5}$ | $\frac{236}{9}$ | $\frac{1795}{14}$ | $\frac{7557}{10}$ |

### [C12] Graphs of $\log Z^{(\lambda)}$
In figure 1 (a), the third black marker
for the value of $\log Z^{(\lambda)}$ seems to be violating
the concavity of $\log Z^{(\lambda)}$ in $\lambda$
assured by Theorem 3.2,
as it seems to be slightly above the line segment
connecting the two adjacent markers,
as can be confirmed by magnifying the figure and applying a ruler.
The same applies to figure 4 (a), figure 5 (a) and figure 7 (a) as well.

### [C13] Section 5.2
The claim that dependence of $\bar{F}^{(\lambda)}$ and $\log Z^{(\lambda)}$
on $\lambda$ is "of a phase transition type" is inappropriate.
Although I know that in some engineering papers the term "phase transition"
is used to merely refer to a rather rapid change of something,
a common convention in statistical mechanics is such that
there should be a definite phase-transition point
where a certain symmetry of the system changes
and/or thermodynamical functions, such as the free energy,
exhibit anomaly/singularity, in the thermodynamic limit.
I could not find any clue for such anomaly in Figures 4-7.
One should not use the term "phase transition"
when what one observes is merely a rapid change.

### [C14] Section 5.3
I do not understand what the authors mean
by the claim "if two or more of empirical estimates
of $Z^{(\lambda)}\mathcal{Z}^{(\lambda)}$ at different $\lambda$ are
sufficiently close to each other we can use them to bound $Z$
from above and below."
As Theorem 4.1 states, $Z^{(\lambda)}\mathcal{Z}^{(\lambda)}$
should be equal to $Z$ irrespectively of the value of $\lambda$,
for $\lambda_1\not=\lambda_2$ one ideally has
$Z^{(\lambda_1)}\mathcal{Z}^{(\lambda_1)}=Z^{(\lambda_2)}\mathcal{Z}^{(\lambda_2)}=Z$.
Thus, if $Z^{(\lambda_1)}\mathcal{Z}^{(\lambda_1)}$
and $Z^{(\lambda_2)}\mathcal{Z}^{(\lambda_2)}$ are sufficiently close
to each other, it might only imply that our optimization
of $\mathcal{B}$ and numerical estimation of $\mathcal{Z}^{(\lambda)}$
are going well.
How can one obtain the claimed bounds of $Z$?

### [C15] Section 5.4
I think that the concentration claim here "the width of the probability
distribution of $\lambda_*$ within the ensemble scales as $\propto1/\sqrt{N}$"
lacks empirical justification.
In Figure 2, only four instances are compared,
and if one compares Figure 2 (b) $20\times20$ and (c) $30\times30$
or (d) $40\times40$,
one can observe that (b) exhibits stronger concentration of $\lambda_*$
(shown by hollow markers)
than (c) or (d), which is in conflict with the authors' concentration claim.

### [C16] Section 5.5
The proposal of the sampling-based
evaluation of $\mathcal{Z}^{(\lambda)}$ was experimented,
and the results were shown in Figure 9 in Appendix E.
They have the following problems:
- Since the results are obtained from random sampling,
  they should have statistical fluctuations.
  Error bars should then be shown to demonstrate
  how large the fluctuations are.
- Figure 9 (a) shows the results
  for the model on the $3\times 3$ grids.
  As the model contains $3\times 3$ binary variables,
  the direct calculation of the partition function $Z$
  will only require $2^{3\times 3}=512$ evaluations
  of the Boltzmann factor and summation of them.
  Therefore, in this particular setting one should say that
  the proposed method is far less efficient than the direct evaluation of $Z$.
- For the setting in Figure 9 (b), direct calculation
  of $Z$ would require $2^{36}\sim 6.9\times 10^{10}$,
  so that the proposed method might seem efficient in terms of
  the number of samples needed. One could however argue that
  alternative sampling-based approaches, such as those
  mentioned in [C5], can be exploited in estimating the partition function $Z$.
  Comparison of the proposed method with such naive approaches
  is needed in order to demonstrate efficiency of the proposal.
- As in [C15], the authors' claim "number of samples needed for
  convergence scales as $\mathcal{O}(N^{[2::4]})$" lacks justification.
  In Figure 9 in Appendix E, the authors show how
  the sampling-based estimates of $\mathcal{Z}^{(\lambda)}$ behave
  as the number $M$ of samples changes.
  They only experimented two cases, $N=3$ and $6$.
  One cannot figure out how the "sufficient number of samples,"
  denoted by $M_c$, behaves as a function of $N$
  on the basis of the results of only two values of $N$.
  Indeed, the following formulas reproduce
  the values of $M_c(N)$ at $N=3,6$, and yet
  they exhibit different dependence on $N$:
  The former is polynomial in $N$ since $M_c(N)=\mathcal{O}(N^\alpha)$,
  whereas the latter is exponential in $N$ since $M_c(N)=\mathcal{O}(e^{cN})$.
  \begin{equation}
    M_c(N)
    \approx(M_c(6)-M_c(3))\left(\frac{N-3}{3}\right)^\alpha+M_c(3),
    \quad\alpha>0,
\end{equation}
\begin{equation}
    M_c(N)\approx M_c(3)\left(\frac{M_c(6)}{M_c(3)}\right)^{(N-3)/3}.
  \end{equation}
  As we do not have any preference among the two candidates as well as
  several other ones, we cannot decide which functional form
  is appropriate for $M_c$.
  The expression $[2::4]$ in the exponent is also ambiguous.
  One may alternatively write something like
  $\mathcal{O}(N^\alpha)$ with $\alpha\in[2,4]$
  if it is a valid statement.
- In page 3, lines 23-24, the authors claim that ``the number of
  samples required for the estimate is either independent on (→ of)
  the system size, $N$, or ...''. I do not find, however, any
  experimental results supporting this claim either in Section 5.5
  or in Appendix E.

### [C16] Minor points
- Page 1, lines 7 and 13; page 3, lines 4 and 5; page 6, line 31; page 7, line 2 and footnote 1: Closing double quotes are typeset differently from opening double quotes.
- Page 1, line 37: there( )in.
- Page 3, line 15: The symbol $\bar{\lambda}$ has not been defined yet.
- Page 3, line 21: suggest(s)
- Page 3, line 24: we(e → a)kly
- Page 3, line 29: finding(,) $\lambda_*\in[0,1]$
- Page 3, line 34: high-dimensional probability distribution(s)
- Page 4, equation (4): $E_{ab}$ → $E_{ab}(x_a,x_b)$ / I feel that the factor $(1/2)$ of the terms
  $h_ax_a+h_bx_b$ is somehow strange, since these terms are to be summed up
  in equation (3) to yield $(1/2)\sum_{a\in\mathcal{V}}d_ah_ax_x$,
  where $d_a$ is the degree of the node $a$ in the graph,
  which may be different from 2.
- Page 4, line 8: which (is), generally, requires
- Page 4, line 11: $\mathcal{B}(x)\in\\\{-1,1\\\}^{|\mathcal{V}|}$
  → $\mathcal{B}:\\\{-1,1\\\}^{|\mathcal{V}|}\to[0,1]$
- Page 4, line 29, equation (9): $\lambda\in[0,1](.\ \to\ ,)$
- Page 5, line 14: in (the → a) lower bound
- Page 6, line 1: line(a)r; ther(e)fore
- Page 6, line 33: from a to b → from $a$ to $b$
- Page 8, lines 5-6: "equally spaaced with the increment 0.05,
  between 0.01 and 1" In order to divide the interval $[0.01,1]$
  evenly, the increment should not be 0.05 but $0.99/20=0.0495$.
- Page 10, line 12: in the lat(t)er case
- Page 12, line 20: The publication year of Wainweight and Jordan is not 2007 but 2008.
- Page 13, line 15: result(s) in
- Page 13, line 18, equation (22): $\eta_{b\to a}^{(\lambda)}(x_a)$
  and $\eta_{a\to b}^{(\lambda)}(x_b)$ on the right-hand side are overlapped.
- Page 14, line 2: from node-to-node → from node to node
- Page 14, equations (24) and (25): Avoid use of the same symbols
  as the independent variables $x_a$ and $x_b$ in the normalization
  coefficients.
- Figure numbers should not be enclosed in pairs of parentheses.

**Strengths And Weaknesses:**

I think that the proposal of extending the Bethe and TRW approximations
is itself not novel (see [C1] below) but interesting, although one should note
that there are several errors and flaws even in the theoretical part
of this paper (see [C3], [C6], [C7], [C9]-[C11]).
I do not precisely understand the proposed algorithm for evaluating $Z$,
because of the unclear description in this paper (see [C4]),
but if my understanding is appropriate,
then one can question the efficiency of the algorithm (see [C5]).
One can furthermore raise several criticisms on
the experimental part of this paper (see [C12]-[C16]).

---

> ### Author Response · Authors · 2024-07-31
> **Responce to Reviewer Nu2b**
>
> ### [C1] Novelity
> We appreciate the reviewer's detailed analysis and the references to related works by Wainwright, Jaakkola, and Willsky (2005), Wainwright and Jordan (2008), and Wiegerinck and Heskes (2003). As pointed out, our formulation indeed builds upon these and number of other foundational works. We discuss it in details in the Introduction.
> In this context, we would like to emphasize that our contribution lies in extending ideas from the papers through a unified framework of fractional interpolation (equations (8)-(9)), where we explicitly introduce the parameter $\lambda$ to interpolate between the TRW and Bethe approximations. This interpolation is not just a simple linear combination but is derived through a rigorous variational framework, guaranteeing bounds and continuity properties that are novel to our approach. Furthermore, our theoretical analysis includes proving the existence of a unique $\lambda^*$ that provides the exact partition function, as well as practical algorithms for computing this parameter.
>
> Explicit citations and discussions in our manuscript to clarify these relationships and better position our contributions in the context of these prior works were carefully checked and added to the current version of our manuscript.appreciate
> ### [C2] Further possible extension
> We appreciate all the suggestions provided by the reviewer. The additional statements suggested by the reviewer appear to be correct and valuable extensions to our work. However, we believe that the paper is already quite dense with information. Our suggestion would be, once our paper is published, for the reviewer to consider publishing these extensions separately.
>
> Concerning connections to, but also differences from, the work discussed in Wiegerinck and Heskes (2003): The connections and relations are discussed in the introductory part of the manuscript and also in our response to the previous comment of the reviewer. Regarding the differences, a major distinction is that Wiegerinck and Heskes did not derive the correction factor to the fractional BP approximation for the partition function. Our work addresses this gap by establishing the relationship between the fractional BP and the exact expression for the partition function, thus providing a more comprehensive framework.
>
> ### [C3] Validity of Theorem 3.1 and Lemma 3.3
> We are thankful to the Referee for helping us identify an additional condition needed to make Theorem 3.1 correct. Specifically, we need to assume that ${\cal B}^{(\lambda)}$ is continuous in $\lambda$ (or equivalently, that $d{\cal B}^{(\lambda)}/d\lambda$ is bounded). Although the possibility of a jump in ${\cal B}^{(\lambda)}$ seems very exotic, we cannot entirely exclude it. We have modified the statement of the Theorem accordingly and added a remark to clarify why this condition is necessary: to prevent the optimal ${\cal  B}^{(\lambda)}$ from jumping at a particular $\lambda$.
>
> ### [C4] Fractional message passing algorithm
> *  We agree with the reviewer. We have corrected the name of the algorithm to Fractional Belief Propagation (FBP).
> *  In our experiments, we follow the numerical approach outlined below: first, we identify the tolerance interval where $\lambda_*$ resides; second, we provide a numerical estimate for $\lambda_*$ at the midpoint of the interval; third, we estimate how the tolerance interval behaves with the number of samples and the system size. Additional discussions and clarifications have been added to Section 5 of the revised manuscript.
>
> ### [C5] Possible alternatives
> Thank you very much for your suggestion. In response, we have introduced Algorithm 2 in the new version of the paper, based on TRW+sampling. We have also compared its performance with the scenario where the exact value $\lambda_*$ is known. For detailed discussion see section 5.5.
>
> ### [C6] Equation (5)
> Thanks a lot for catching the typo. Corrected.
>
> ### [C7] Derivation of fractional free energy
> The referee is right; our notations were not properly explained and were confusing. The definition of ${\cal  B}$ as a union of node and edge marginals, exactly as suggested by the referee, is given in the text just before Eq.~(8). We have introduced the definition of ${\cal  B}^{(\lambda)} = \text{arg}\min_{{\cal B}} F^{(\lambda)}(\mathcal{B})$ and clarified that $\bar{F}^{(\lambda)}$ represents the value of the fractional free energy at the minimum. Additionally, we have replaced the $(a,b)$ notations with {$a,b$} to denote edges of the graph consistently throughout the paper.
>
> ### [C8] Theorem 3.2
> Thanks a lot for the comment. We added suggested proof to the paper.
>
> ### [C9] Lemma 3.3
> We have added a sentence to the proof to clarify the attractive property more explicitly.

---

> > ### Author Response · Authors · 2024-07-31
> > **Responce to Reviewer Nu2b**
> >
> > ### [C10] Theorem 4.1
> > We appreciate the referee's insightful comments and agree with the assessment. We would like to clarify the relation between our new results and existing work. Proposition 4.3 in Wainwright and Jordan (2007) introduces Tree-Based Reparameterization for graphical models, which is exact for tree structures and approximate for graphs with cycles. In our paper, we extend this tree-based reparameterization to a more general class. Theorem 4.1 helps us account for the error in this approximation by introducing a correction factor. While our method builds upon the tree-based reparameterization framework, Theorem 4.1 is a novel contribution that specifically addresses the effects of this approximation in more complex graph structures.
> >
> > Furthermore, we are thankful to the referee for identifying the typo in equation (28) in Appendix C. We have corrected the factor in the revised manuscript.
> >
> > ### [C11] Lemma 5.1
> > We greatly appreciate the referee's detailed analysis, which highlighted issues in Lemma 5.1 and Algorithm 2 (in the original version). Guided by the referee's remarks, we recognized that edge-uniform weights are not always feasible. This was briefly mentioned in passing in [Wainwright PhD thesis, 2002], but without proper emphasis. Consequently, we have decided to remove Lemma 5.1 and the respective Appendix from the text. These sections were originally included as a side comment, which, as the referee correctly pointed out, is neither accurate nor relevant.
> >
> > ### [C12] Graphs of $\log Z^{(\lambda)}$
> >  Thank you very much. The issue noted by the referee was due to a bug in our code. With the bug corrected, we have updated all the figures in the paper.
> > ### [C13] Section 5.2
> >  With the bug identified (thanks again to the referee), we discovered that the abrupt transition does not hold in at least some cases. Therefore, we have removed the phase transition claim from the latest version of the paper.
> >
> > ### [C14]  Section 5.3
> > The reviewer is correct; the statement is indeed confusing. Therefore, we have decided to remove it from the paper.
> >
> > ### [C15]  Section 5.4
> >  We updated the corresponding figure and report $\frac{\log Z^{(\lambda)}}{N}$. In this case as $N\to \infty$ the concentration behavior is concluded from the figures.
> >
> > ### [C16]  Section 5.5
> > *  Fig. 3 reports the dependence of the sample-based estimate of $Z$ on the number of samples and $\lambda$ where we performed the experiment for 100 times and report the mean and variance. We observe the dependence of number of samples on $\lambda$. If $\lambda = \lambda_*$  number of samples to estimate the partition function is fewer than $\lambda = 0$.   Our major observation here is that the result converges with an increase in the number of samples. Moreover, comparing the speed of convergence to the size of the system, $N$, we estimate that the number of samples needed for convergence scales as $\mathcal{O}(N^{4})$.
> > * In Fig.~3, we compared two cases when $\lambda =0$ and $\lambda = \lambda_*$. In the latter when exact value is known, fewer samples is required.
> >
> > * For the last comment,  we are dropping the claim.
> > ### [C16] Minor points
> > Thanks a lot!  We have corrected all the misprints spotted by the referee.

---

### Author Response · Authors · 2024-07-31
**Updated version of our manuscript is now available**

Dear Editor,

We are grateful for the opportunity to revise our manuscript and appreciate your selection of such an excellent group of referees. All the comments and suggestions provided by the three referees were highly appropriate and insightful.

We believe that we have thoroughly addressed all the comments in the revised manuscript. We would like to extend our special thanks to Reviewer Nu2b for the exceptionally high-quality report.

---

> ### Comment · Reviewer_NNDJ · 2024-08-20
>
> The new manuscript is indeed much improved. There are however still some serious issues, that I list here in order of importance. The first two the main issues, which in my opinion must be resolved if the paper is to be accepted.
>
> ## 1. Missing clear statement of actual algorithm
> Algorithm 1 as written is a bit confusing: because of Theorem 4.1 one can just pick an arbitrary $\lambda \in [0,1]$ and use that to estimate $Z$ as the product of $Z^{(\lambda)}$ and $\tilde{Z}^{(\lambda)̊}$. From Algorithm 1 itself it is not clear what the benefit is of finding $\lambda_*$ is.
>
> If I understand correctly, the practical use of FBP would be to use a lot of computational resources to precompute $\lambda_*$ for a certain ensemble of GMs, and then use it to quickly estimate the the free energy or marginals when presented with a new instances of this GM. If my understanding is correct I think this should also be stated formally as an algorithm. If I have misunderstood, then the presentation of the algorithm is insufficient.
>
> ## 2. Too strong claim in Section 5.5
> Figure 3 delivers empirical evidence that the estimates of FBP at $\lambda=\lambda_*$ converge faster and have a lower variance than FBP at $\lambda=0$, a.k.a. TRW. But it seems like a stretch to read off specific exponents like $O(N^4)$ and $O(N^2)$ from this figure. I would phrase the latter as plausible guesses based on the experiment, not solid empirical results. Maybe this was the intention of the authors when saying "we estimate" - if so the wording needs to be reworked to emphasize the uncertainty in these "estimates" (for instance the sentence "However, the scaling becomes much better, $O(N^2)$...." reads like the statement of a definitive claim).
>
> ## 3. Section 5.7
> I don't understand the optimization of the parameter $J$. First of all, why it is done at all is not clear to me. Secondly, the "error" defined in the display equation seems like it should be a function only of the noise level $\varepsilon$, not of $J$. Should it be "different pixels in true image and *de*noised version'"?
>
> ## 4. Section 6
> What is meant by "extrapolation" in the last two bullet points? This is the first time the word is used in the manuscript, and it is not clear to me what it refers to.
>
> ## 5. Abstract
> The notation 2::4 has not been introduced yet.
>
> ## 6. Theorem 4.1
> The statement should have a qualifier, probably "For any  $\lambda \in [0,1]$".
>
> ## 7. p8
> The sentence starting with "A direct corollary of Lemma..."  is a bit convoluted. I would make the part in the paranthesis an independent sentence.
>
> ## 8. p8
> The sentence starting with "We introduce the $\lambda$-optimal FBP....". This sentence reads a bit strange, since the algorithms have already been introduced. Past tense would be more appropriate.
>
> ## 9. Language and grammar
> The manuscript contains many grammatical and typographical errors (missing or incorrect punctuation, missing or erroneous definitve articles, incorrect capitalization, etc). I have highlighted many such errors in a PDF that I will ask the editor to pass on to the authors. Unfortunately it does not contain explanations of what is wrong, but ChaptGPT or similar should be able to explain.

---

> > ### Author Response · Authors · 2024-10-15
> > **Responce to Reviewer NNDJ**
> >
> > **1.Missing clear statement of actual algorithm**
> >
> > Thank you for your insightful comments. We appreciate the opportunity to clarify our approach and have made the following improvements to address your concerns:
> > 1. Importance of finding $\lambda^*$:
> > While it's true that we can choose an arbitrary $\lambda \in (0,1)$ to estimate $Z$ as per Theorem 4.1, finding $\lambda^*$ offers significant advantages. As discussed in Section 5.5 and illustrated in Fig. 3, when $\lambda$ is not close to $\lambda^*$, we require more samples to accurately estimate the free energy. By finding $\lambda^*$, we can achieve higher accuracy with fewer samples.
> >
> > 2. Practical use of FBP for large ensembles:
> > Your understanding is correct, and we thank you for highlighting the need for a clearer presentation of this practical application. To address this, we have added Algorithm 3, which formally states the process for large ensembles.
> >
> >
> > We believe these changes and additions significantly improve the clarity and practical relevance of our presentation. The new Algorithm 3 provides a formal framework for applying FBP to large ensembles, addressing your concern about the practical use of the method.
> >
> >
> >
> >
> >
> >
> > **2. Too strong claim in Section 5.5**
> >
> > We appreciate the reviewer's feedback and agree with the suggestion. To address the concern, we have revised the wording in Section 5.5 to reflect the speculative nature of the scaling behavior. Specifically, we have replaced "estimate" with "conjecture" to better emphasize the uncertainty in these observations. We have also adjusted the phrasing in other relevant sentences to ensure they do not imply definitive claims, but rather plausible conjectures based on the experimental data.
> >
> >
> > **3. Section 5.7**
> >
> > Regarding the Reviewer's first point: \( $\mathbf{J}$ \) represents the degree of similarity between nearby pixels. By adjusting  \( $\mathbf{J}$ \), we can explore different denoising regimes. In all our experiments, the level of noise is fixed, so the dependence of the ratio on \( $\varepsilon$ \) is not a concern. Secondly, we appreciate the reviewer for pointing out the typo—it should indeed be the "denoised version." This has been corrected in the revised manuscript.
> >
> >
> >
> > **4. Section 6**
> >
> > We appreciate the reviewer’s observation and agree that the term "extrapolation" was used incorrectly. To clarify, we have replaced "extrapolation" with "interpolation" in the last two bullet points. The revised wording better reflects the intended meaning and improves consistency throughout the manuscript.
> >
> >
> >
> > **5. Abstract**
> >
> > Thank you for pointing this out. We have now introduced the notation "2::4" in the abstract to clarify its meaning and ensure that readers can interpret it correctly from the outset.
> >
> >
> > **6. Theorem 4.1**
> >
> > We thank the reviewer for a good suggestion -- the qualifier is added.
> >
> > **7. p8**
> >
> > We agree with the reviewer’s suggestion. To improve clarity and avoid confusion, we have divided the sentence into two, making the part in parentheses a separate, independent sentence.
> >
> >
> > **8. p8**
> >
> > We agree with the reviewer. To improve readability, we have revised the sentence by using the past tense, ensuring consistency with the earlier introduction of the algorithms.
> >
> > **9. Language and grammar**
> >
> > We thank the Reviewer for sharing the annotated PDF. We carefully addressed all the highlighted language and grammar issues. The feedback is greatly appreciated.

---

### Author Response · Authors · 2024-10-15
**New Update**

Dear Editor,

We are pleased to submit our response to the latest comments from Reviewer NNDJ. We have carefully addressed all the concerns raised.

Thank you for your continued consideration.

---

> ### Comment · Reviewer_Nu2b · 2024-10-22
>
> I have read the revised manuscript posted on October 16, which I found still has several flaws, both theoretically and experimentally, which would require significant revision.
>
> ### [C1] Theorem 3.1
> It assumes continuity of $\mathcal{B}^{(\lambda)}$.
> - In the remark that follows the theorem, the authors claim that the continuity almost always holds. I would also expect that it be the case, but from the theoretical point of view it is not certain whether such an expectation holds true.
> - In the proof in Appendix B, the authors consider the derivatives of $\mathcal{B}_{ab}^{(\lambda)}$ and $\mathcal{B}_a^{(\lambda)}$
> with respect to $\lambda$, while the continuity assumption alone does not allow us to take the derivatives. It makes the proof invalid.
>
> ### [C2] Alternative reasoning justifying existence of $\lambda_*$
> I think that existence of $\lambda_*$ claimed in Lemma 3.3 might be proved without the continuity assumption of $\mathcal{B}^{(\lambda)}$, as follows. Consider the extended formulation that I mentioned in [C2] of my initial review comments. Then the same argument as in the proof of Theorem 3.2 proves that $\bar{F}^{(\rho)}$ is concave in $\rho$ on $[0,\infty)^{|\mathcal{E}|}$. In particular, $\bar{F}^{(\rho)}$ is concave on an open set containing the line segment connecting the values of $\rho$ corresponding to the TRW and BP, both residing in the interior of $[0,\infty)^{|\mathcal{E}|}$. The concavity in turn implies the continuity, and the existence of $\lambda_*$ then follows from the intermediate value theorem.
>
> The following points should be noted:
> - The concavity of $\bar{F}^{(\lambda)}$ on the interval $\lambda\in[0,1]$ is not enough to claim the continuity of $\bar{F}^{(\lambda)}$ on $[0,1]$, because the concavity does not constrain the values of $\bar{F}^{(0)}$ and $\bar{F}^{(1)}$ enough to guarantee the desired continuity including the endpoints of $[0,1]$.
> - The set of points where a concave function is non-differentiable can be dense. It makes the proof of the monotonicity based on the derivatives given in Appendix B unjustified, at least in its current form.
> - It seems that the concavity alone does not guarantee the uniqueness of $\lambda_*$.
>
> ### [C3] Algorithm 1
> The problem that I raised in the second item of [C4] in my initial review still persists. A more concrete description should be added.
> > The authors write "Find $\lambda_*$ where $\tilde{\mathcal{Z}}^{(\lambda)}=1$", but how?
> According to Algorithm 1, one estimates the values of $\tilde{\mathcal{Z}}^{(\lambda)}$ via random sampling for 21 equispaced values of
> $\lambda$ from 0 to 1. Among the resulting 21 estimates $\tilde{\mathcal{Z}}^{(\lambda=0)},\ldots,\tilde{\mathcal{Z}}^{(\lambda=1)}$, it is quite unlikely that there exists a value of $\lambda$ with which the value of $\tilde{\mathcal{Z}}^{(\lambda)}$ is estimated to be exactly 1. One should also take care of the statistical fluctuations of these estimates arising from the random sampling.
>
> ### [C4] Algorithm 2
> - I did not understand the significance of Algorithm 2. First of all, the objective of this paper is to propose an algorithm to compute $Z$. If the exact value of $Z$ were known as an input to Algorithm 2, one does not have to do anything. It also seems that Algorithm 2 was not used in the experiments in this paper at all.
> - According to the description in the main text, it should return the value of $\lambda_*$.
>
> ### [C5] Section 5.4
> - It does not make sense to compare variability in the value of $\bar{F}^{(\lambda)}$ and that in $\lambda_*$, because the nominal variability depends on the scale of the quantity of interest.
> - In the first paragraph, why did the authors argue solely on the basis of Figure 5? Figure 2 would also provide experimental results related to the argument there.
> - The claim on the *width of the distribution* of $\lambda_*$ being proportional to $1/\sqrt{N}$ is not justifiable. The authors made this claim solely on the basis of the experimental results shown in Figure 2, which are on a single model (planar zero-field Ising) with only four different values of $N$, with only four samples for each value of $N$, and without specifying how the *width* is to be defined. This setting is far from sufficient to draw any quantitative statement like "$\propto 1/\sqrt{N}$".
> - Algorithm 3 is not well organized. It only shows definition of the two functions PrecomputeLambdaStar() and ComputePartitionAngMarginals(), and does not show how one combines them to execute inference on $G_{\rm new}$. I can guess it, but it should be described clearly. I did not understand either the sentence "This is also indicated in Algorithm 3" in the main text. What the objective of Algorithm 3 is should be stated explicitly.

---

> ### Comment · Reviewer_Nu2b · 2024-10-22
> **Official Comment by Reviewer Nu2b (continued)**
>
> ### [C6] Section 5.5
> I did not understand the claims on the scalings such as $\mathcal{O}(N^4),\mathcal{O}(N^2)$, etc. Upon reading the $\mathcal{O}(N^4)$ claim, I presumed that it say that at most the number of samples that is *proportional to $N^4$* is needed to obtain a reliable estimate of $\log Z$. It is not the same as claiming that $N^4$ samples are enough, as implied by Figure 3.
>
> Also, when discussing the sample complexity, the authors should take into account not only the number of samples needed to estimate $\log Z$ but also the number of samples needed to determine the value of $\lambda_*$.
>
> ### [C7] Section 5.7
> The image denoising experiment has the following problem. In determining the model parameter $J$ the authors use the true image. This assumption of the availability of the true image makes the proposal not usable in practice.

---

### Decision · Action_Editor_oQGR · 2024-10-30

**Recommendation:** Accept as is

**Comment:**

After careful review of the author’s responses and revisions, I recommend accepting the paper. The submission introduces a valuable interpolation approach between BP and TRW, offering new insights into inference for structured models. Reviewer concerns on theoretical assumptions and algorithmic clarity were addressed, and additional empirical validation has strengthened the submission. Although some empirical claims remain conjectural, the core contributions are well-supported.

**Audience:**

This paper addresses a relevant challenge in graphical model inference, with contributions that will appeal to TMLR’s audience, especially those in probabilistic inference and message-passing algorithms. Its findings could be particularly valuable to researchers focused on structured model inference in both statistical physics and machine learning contexts.

**Claims And Evidence:**

The paper studies a novel interpolation framework between Tree-Reweighted Belief Propagation (TRW-BP) and standard Belief Propagation (BP) for partition function approximation in Ising models, a well-studied and classical example of graphical model. The authors support their claims with rigorous derivations and algorithmic implementations, specifically proving key properties like continuity and factorization. Reviewer feedback on continuity assumptions and uniqueness of "lambda" seem to have been addressed through revisions and additional empirical evidence. Though focused on synthetic data, the experiments validate the algorithmic claims.